# STRENGTH OF MINIBATCH NOISE IN SGD

**Liu Ziyin**[*]**, Kangqiao Liu**[*]**, Takashi Mori, & Masahito Ueda**
The University of Tokyo

## ABSTRACT

The noise in stochastic gradient descent (SGD), caused by minibatch sampling, is poorly understood despite its practical importance in deep learning. This work presents the first systematic study of the SGD noise and fluctuations close to a local minimum. We first analyze the SGD noise in linear regression in detail and then derive a general formula for approximating SGD noise in different types of minima. For application, our results (1) provide insight into the stability of training a neural network, (2) suggest that a large learning rate can help generalization by introducing an implicit regularization, (3) explain why the linear learning rate-batchsize scaling law fails at a large learning rate or at a small batchsize and (4) can provide an understanding of how discrete-time nature of SGD affects the recently discovered power-law phenomenon of SGD.

## 1 INTRODUCTION

Stochastic gradient descent (SGD) is the simple and efficient optimization algorithm behind the success of deep learning (Allen-Zhu et al., 2019; Xing et al., 2018; Zhang et al., 2018; Wang et al., 2020; He and Tao, 2020; Liu et al., 2021; Simsekli et al., 2019; Wu et al., 2020). Minibatch noise, also known as the SGD noise, is the primary type of noise in the learning dynamics of neural networks. Practically, minibatch noise is unavoidable because a modern computer's memory is limited while the size of the datasets we use is large; this demands the dataset to be split into "minibatches" for training. At the same time, using minibatch is also a recommended practice because using a smaller batch size often leads to better generalization performance (Hoffer et al., 2017). Therefore, understanding minibatch noise in SGD has been one of the primary topics in deep learning theory. Dominantly many theoretical studies take two approximations: (1) the continuous-time approximation, which takes the infinitesimal step-size limit; (2) the Hessian approximation, which assumes that the covariance matrix of the SGD noise is equal to the Hessian $H$. While these approximations have been shown to provide some qualitative understanding, the limitation of these approximations is not well understood. For example, it is still unsure when such approximations are valid, which hinders our capability to assess the correctness of the results obtained by approximations.

In this work, we fill this gap by deriving analytical formulae for discrete-time SGD with arbitrary learning rates and exact minibatch noise covariance. In summary, the main **contributions** are: (1) we derive the strength and the shape of the minibatch SGD noise in the cases where the noise for discrete-time SGD is analytically solvable; (2) we show that the SGD noise takes a different form in different kinds of minima and propose general and more accurate approximations. This work is organized as follows: Sec. 2 introduces the background. Sec. 3 discusses the related works. Sec. 4 outlines our theoretical results. Sec. 5 derives new approximation formulae for SGD noises. In Sec. 6, we show how our results can provide practical and theoretical insights to problems relevant to contemporary machine learning research. For reference, the relationship of this work to the previous works is shown in Table 1.

## 2 BACKGROUND

In this section, we introduce the minibatch SGD algorithm. Let $\{x_i, y_i\}_{i=1}^N$ be a training set. We can define the gradient descent (GD) algorithm for a differentiable loss function $L$ as $\mathbf{w}_t = \mathbf{w}_{t-1} - \lambda \nabla_{\mathbf{w}} L(\mathbf{w}, \{\mathbf{x}, \mathbf{y}\})$, where $\lambda$ is the learning rate and $\mathbf{w} \in \mathbb{R}^D$ is the weights of the model. We consider an additive loss function for applying the minibatch SGD.

**Definition 1.** A loss function $L(\{x_i, y_i\}_{i=1}^N, \mathbf{w})$ is additive if $L(\{x_i, y_i\}_{i=1}^N, \mathbf{w}) = \frac{1}{N} \sum_{i=1}^N \ell(x_i, y_i, \mathbf{w})$ for some differentiable, non-negative function $\ell(\cdot)$.

Table 1: Summary of related works on the noise and stationary distribution of SGD. This work fills the gap of the lack of theoretical results for the actual SGD dynamics, which is *discrete-time* and with *minibatch noise*.

| Setting | Artificial Noise | Hessian Approximation Noise | Minibatch Noise |
|---|---|---|---|
| Continuous-time | Sato and Nakagawa (2014); Welling and Teh (2011) Mandt et al. (2017); Meng et al. (2020) | Jastrzebski et al. (2018); Zhu et al. (2019) Wu et al. (2020); Xie et al. (2021) | Blanc et al. (2020); Mori et al. (2021) |
| *Discrete-time* | Yaida (2019); Gitman et al. (2019) Liu et al. (2021) | Liu et al. (2021) | **This Work** |

This definition is quite general. Most commonly studied and used loss functions are additive, e.g., the mean-square error (MSE) and cross-entropy loss. For an additive loss, the minibatch SGD with momentum algorithm can be defined.

**Definition 2.** The *minibatch SGD with momentum* algorithm by sampling with replacement computes the update to the parameter $\mathbf{w}$ with the following set of equations:

$$\begin{cases} \hat{\mathbf{g}}_t = \frac{1}{S} \sum_{i \in B_t} \nabla \ell(x_i, y_i, \mathbf{w}_{t-1}); \\ \mathbf{m}_t = \mu \mathbf{m}_{t-1} + \hat{\mathbf{g}}_t; \\ \mathbf{w}_t = \mathbf{w}_{t-1} - \lambda \mathbf{m}_t. \end{cases} \tag{1}$$

where $\mu \in [0,1)$ is the momentum hyperparameter, $S := |B_t|$ is the minibatch size, and the set $B_t = \{i_1, ... i_S\}$ are $S$ i.i.d. random integers sampled uniformly from $[1, N]$.

One can decompose the gradient into a deterministic plus a stochastic term. Note that $\mathbb{E}_B[\hat{\mathbf{g}}_t] = \nabla L$ is equal to the gradient for the GD algorithm. We use $\mathbb{E}_B(\cdot)$ to denote the expectation over batches, and use $\mathbb{E}_{\mathbf{w}}(\cdot)$ to denote the expectation over the stationary distribution of the model parameters. Therefore, we can write $\hat{\mathbf{g}}_t = \mathbb{E}_B[\hat{\mathbf{g}}_t] + \eta_t$, where $\eta_t := \frac{1}{S} \sum_{i \in B_t} \nabla \ell(x_i, y_i, \mathbf{w}_{t-1}) - \mathbb{E}_B[\hat{\mathbf{g}}_t]$ is the noise term; the noise covariance is $C(\mathbf{w}_t) := \text{cov}(\eta_t, \eta_t)$. Of central importance to us is the averaged asymptotic noise covariance $C := \lim_{t \to \infty} \mathbb{E}_{\mathbf{w}_t}[C(\mathbf{w}_t)]$. Also, we consider the asymptotic model fluctuation $\Sigma := \lim_{t \to \infty} \text{cov}(\mathbf{w}_t, \mathbf{w}_t)$. $\Sigma$ gives the strength and shape of the fluctuation of $\mathbf{w}$ around a local minimum and is another quantity of central importance to this work. Throughout this work, $C$ is called the "noise" and $\Sigma$ the "fluctuation".

# 3 RELATED WORKS

**Noise and Fluctuation in SGD**. Deep learning models are trained with SGD and its variants. To understand the parameter distribution in deep learning, one needs to understand the stationary distribution of SGD (Mandt et al., 2017). Sato and Nakagawa (2014) describes the stationary distribution of stochastic gradient Langevin dynamics using discrete-time Fokker-Planck equation. Yaida (2019) connects the covariance of parameter $\Sigma$ to that of the noise $C$ through the fluctuation-dissipation theorem. When $\Sigma$ is known, one may obtain by Laplace approximation the stationary distribution of the model parameter around a local minimum $w^*$ as $\mathcal{N}(w^*, \Sigma)$. Therefore, knowing $\Sigma$ can be of great practical use. For example, it has been used to estimate the local minimum escape efficiency (Zhu et al., 2019; Liu et al., 2021) and argue that SGD prefers a flatter minimum; it can also be used to assess parameter uncertainty and prediction uncertainty when a Bayesian prior is specified (Mandt et al., 2017; Gal and Ghahramani, 2016; Pearce et al., 2020). Empirically, both the fluctuation and the noise are known to crucially affect the generalization of a deep neural network. Wu et al. (2020) shows that the strength and shape of the $\Sigma$ due to the minibatch noise lead to better generalization of neural networks in comparison to an artificially constructed noise.

**Hessian Approximation of the Minibatch Noise.** However, it is not yet known what form $C$ and $\Sigma$ actually take for SGD in a realistic learning setting. Early attempts assume isotropic noise in the continuous-time limit (Sato and Nakagawa, 2014; Mandt et al., 2017). In this setting, the noise is an isotropic Gaussian with $C \sim I_D$, and $\Sigma$ is known to be proportional to the inverse Hessian $H^{-1}$. More recently, the importance of noise structure was realized (Hoffer et al., 2017; Jastrzebski et al., 2018; Zhu et al., 2019; HaoChen et al., 2020). "Hessian approximation", which assumes $C \approx c_0 H$ for some unknown constant $c_0$, has often been adopted for understanding SGD (see Table 1); this assumption is often motivated by the fact that $C = J_w \approx H$, where $J_w$ is the Fisher information matrix (FIM) (Zhu et al., 2019); the fluctuation can be solved to be isotropic: $\Sigma \sim I_D$. However, it is not known under what conditions the Hessian approximation is valid, while previous works have argued that it can be very inaccurate (Martens, 2014; Liu et al., 2021; Thomas et al., 2020; Kunstner et al., 2019). However, Martens (2014) and Kunstner et al. (2019) only focuses on the natural gradient descent (NGD) setting; Thomas et al. (2020) is closest to ours, but it does not apply to the case with momentum, a matrix learning rate, or regularization.

**Discrete-time SGD with a Large Learning Rate**. Recently, it has been realized that networks trained at a large learning rate have a dramatically better performance than networks trained with a vanishing learning rate (lazy training) (Chizat and Bach, 2018). Lewkowycz et al. (2020) shows that there is a qualitative difference between the lazy training regime and the large learning rate regime; the performance features two plateaus in testing accuracy in the two regimes, with the large learning rate regime performing much better. However, the theory regarding discrete-time SGD at a large learning rate is almost non-existent, and it is also not known what $\Sigma$ may be when the learning rate is non-vanishing. Our work also sheds light on the behavior of SGD at a large learning rate. Some other works also consider discrete-time SGD in a similar setting (Fontaine et al., 2021; Dieuleveut et al., 2020; Toulis et al., 2017), but the focus is not on deriving analytical formulae or does not deal with the stationary distribution.

## 4 SGD NOISE AND FLUCTUATION IN LINEAR REGRESSION

This section derives the shape and strength of SGD noise and fluctuation for linear regression; concurrent to our work, Kunin et al. (2021) also studies the same problem but with continuous-time approximation; our result is thus more general. To emphasize the message, we discuss the label noise case in more detail. The other situations also deserve detailed analysis; we delay such discussion to the appendix due to space constraints. **Notation**: $S$ denotes the minibatch size. $\mathbf{w} \in \mathbb{R}^D$ is the model parameter viewed in a vectorized form; $\lambda \in \mathbb{R}_+$ denotes a scalar learning rate; when the learning rate takes the form of a preconditioning matrix, we use $\Lambda \in \mathbb{R}^{D \times D}$. $A \in \mathbb{R}^{D \times D}$ denotes the covariance matrix of the input data. When a matrix $X$ is positive semi-definite, we write $X \geq 0$; throughout, we require $\Lambda \geq 0$. $\gamma \in \mathbb{R}$ denotes the weight decay hyperparameter; when the weight decay hyperparameter is a matrix, we write $\Gamma \in \mathbb{R}^{D \times D}$. $\mu$ is the momentum hyperparameter in SGD. For two matrices $X, Y$, the commutator is defined as $[X, Y] := XY - YX$. Other notations are introduced in the context.[1] The results of this section are numerically verified in Appendix A.

### 4.1 KEY PREVIOUS RESULTS

When $N \gg S$, the following proposition is well-known and gives the exact noise due to minibatch sampling. See Appendix E.1 for a derivation.

**Proposition 1.** *The noise covariance of SGD as defined in Definition 2 is*

$$C(\mathbf{w}) = \frac{1}{SN} \sum_{i=1}^{N} \nabla \ell_i(\mathbf{w}) \nabla \ell_i(\mathbf{w})^{\mathrm{T}} - \frac{1}{S} \nabla L(\mathbf{w}) \nabla L(\mathbf{w})^{\mathrm{T}}, \tag{2}$$

*where the notations $\ell_i(\mathbf{w}) := l(x_i, y_i, \mathbf{w})$ and $L(\mathbf{w}) := L(\{x_i, y_i\}_{i=1}^{N}, \mathbf{w})$ are used.*

This gradient covariance matrix $C$ is crucial to understand the minibatch noise. The standard literature often assumes $C(\mathbf{w}) \approx H(\mathbf{w})$; however, the following well-known proposition shows that this approximation can easily break down.

**Proposition 2.** *Let $\mathbf{w}_*$ be the solution such that $L(\mathbf{w}_*) = 0$, then $C(\mathbf{w}_*) = 0$.*

*Proof.* Because $\ell_i$ is non-negative for all $i$, $L(\mathbf{w}_*) = 0$ implies that $\ell_i(\mathbf{w}_*) = 0$. The differentiability in turn implies that each $\nabla \ell_i(\mathbf{w}_*) = 0$; therefore, $C = 0$. □

This proposition implies that there is no noise if our model can achieve zero training loss (which is achievable for an overparametrized model). This already suggests that the Hessian approximation $C \sim H$ is wrong since the Hessian is unlikely to vanish in any minimum. The fact that the noise strength vanishes at $L = 0$ suggests that the SGD noise might at least be proportional to $L(\mathbf{w})$, which we will show to be true for many cases. The following theorem relates $C$ and $\Sigma$ of the discrete-time SGD algorithm with momentum for a matrix learning rate.

**Theorem 1.** *(Liu et al., 2021) Consider running SGD on a quadratic loss function with Hessian $H$, learning rate matrix $\Lambda$, momentum $\mu$. Assuming ergodicity, then*

$$(1 - \mu)(\Lambda H \Sigma + \Sigma H \Lambda) - \frac{1 + \mu^2}{1 - \mu^2} \Lambda H \Sigma H \Lambda + \frac{\mu}{1 - \mu^2}(\Lambda H \Lambda H \Sigma + \Sigma H \Lambda H \Lambda) = \Lambda C \Lambda. \tag{3}$$

Propostion 1 and Theorem 1 allow one to solve $C$ and $\Sigma$. Equation (3) can be seen as a general form of the Lyapunov equation (Lyapunov, 1992) and is hard to solve in general (Hammarling, 1982; Ye et al., 1998; Simoncini, 2016). Solving this analytical equation in settings of machine learning relevance is one of the main technical contributions of this work.

---

[1] We use the word *global minimum* to refer to the global minimum of the loss function, i.e., where $L = 0$ and a *local minimum* refers to a minimum that has a non-negative loss, i.e., $L \geq 0$.

## 4.2 RANDOM NOISE IN THE LABEL

We first consider the case when the labels contain noise. The loss function takes the form

$$L(\mathbf{w}) = \frac{1}{2N} \sum_{i=1}^{N} (\mathbf{w}^{\mathrm{T}} x_i - y_i)^2, \tag{4}$$

where $x_i \in \mathbb{R}^D$ are drawn from a zero-mean Gaussian distribution with feature covariance $A := \mathbb{E}_{\mathrm{B}}[x x^{\mathrm{T}}]$, and $y_i = \mathbf{u}^{\mathrm{T}} x_i + \epsilon_i$, for some fixed $\mathbf{u}$ and $\epsilon_i \in \mathbb{R}$ is drawn from a distribution with zero mean and finite second momentum $\sigma^2$. We redefine $\mathbf{w} - \mathbf{u} \to \mathbf{w}$ and let $N \to \infty$ with $D$ held fixed. The following lemma finds $C$ as a function of $\Sigma$.

**Lemma 1.** (*Covariance matrix for SGD noise in the label*) *Let $N \to \infty$ and the model be updated according to Eq. (1) with loss function in Eq. (4). Then,*

$$C = \frac{1}{S} (A \Sigma A + \mathrm{Tr}[A \Sigma] A + \sigma^2 A). \tag{5}$$

The model fluctuation can be obtained using this lemma.

**Theorem 2.** (*Fluctuation of model parameters with random noise in the label*) *Let the assumptions be the same as in Lemma 1 and $[\Lambda, A] = 0$. Then,*

$$\Sigma = \frac{\sigma^2}{S} \left( 1 + \frac{\kappa_\mu}{S} \right) \Lambda G_\mu^{-1}, \tag{6}$$

*where $\kappa_\mu := \frac{\mathrm{Tr}[\Lambda A G_\mu^{-1}]}{1 - \frac{1}{S} \mathrm{Tr}[\Lambda A G_\mu^{-1}]}$ with $G_\mu := 2(1 - \mu)I_D - \left( \frac{1-\mu}{1+\mu} + \frac{1}{S} \right) \Lambda A$.*

**Remark.** *This result is numerically validated in Appendix A. The subscript $\mu$ refers to momentum. To obtain results for vanilla SGD, one can set $\mu = 0$, which has the effect of reducing $G_\mu \to G = 2I_D - \left( 1 + \frac{1}{S} \right) \Lambda A$. From now on, we focus on the case when $\mu = 0$ for notational simplicity, but we note that the results for momentum can be likewise studied. The assumption $[\Lambda, A] = 0$ is not too strong because this condition holds for a scalar learning rate and common second-order methods such as Newton's method.*

If $\sigma^2 = 0$, then $\Sigma = 0$. This means that when there is no label noise, the model parameter has a vanishing stationary fluctuation, which corroborates Proposition 2. When a scalar learning rate $\lambda \ll 1$ and $1 \ll S$, we have

$$\Sigma \approx \frac{\lambda \sigma^2}{2S} I_D, \tag{7}$$

which is the result one would expect from the continuous-time theory with the Hessian approximation (Liu et al., 2021; Xie et al., 2021; Zhu et al., 2019), except for a correction factor of $\sigma^2$. Therefore, a Hessian approximation fails to account for the randomness in the data of strength $\sigma^2$. We provide a systematic and detailed comparison with the Hessian approximation in Table 2 of Appendix B.

Moreover, it is worth comparing the exact result in Theorem 2 with Eq. (7) in the regime of non-vanishing learning rate and small batch size. One notices two differences: (1) an anisotropic enhancement, appearing in the matrix $G_\mu$ and taking the form $-\lambda(1 + 1/S)A$; compared with the result in Liu et al. (2021), this term is due to the compound effect of using a large learning rate and a small batchsize; (2) an isotropic enhancement term $\kappa$, which causes the overall magnitude of fluctuations to increase; this term does not appear in the previous works that are based on the Hessian approximation and is due to the minibatch sampling process alone. As the numerical example in Appendix A shows, at large batch size, the discrete-time nature of SGD is the leading source of fluctuation; at small batch size, the isotropic enhancement becomes the dominant source of fluctuation. Therefore, the minibatch sampling process causes two different kinds of enhancement to the fluctuation, potentially increasing the exploration power of SGD at initialization but reducing the convergence speed.

Now, combining Theorem 2 and Lemma 1, one can obtain an explicit form of the noise covariance.

**Theorem 3.** *The noise covariance matrix of minibatch SGD with random noise in the label is*

$$C = \frac{\sigma^2}{S} A + \frac{\sigma^2}{S^2} \left( 1 + \frac{\kappa_\mu}{S} \right) \left( \Lambda A G_\mu^{-1} + \mathrm{Tr}[\Lambda A G_\mu^{-1}] I_D \right) A. \tag{8}$$

By definition, $C = J$ is the FIM. The Hessian approximation, in sharp contrast, can only account for the term in orange. A significant modification containing both anisotropic and isotropic (up to Hessian) is required to fully understand SGD noise, even in this simple example. Additionally, comparing this result with the training loss (127), one can find that the noise covariance contains one term that is proportional to the training loss. In fact, we will derive in Sec. 5 that containing a term proportional to training loss is a general feature of the SGD noise. We also study the case when the input is contaminated with noise. Interestingly, the result is the same with the label noise case with $\sigma^2$ replaced by a more complicated term of the form $\text{Tr}[AK^{-1}BU]$. We thus omit this part from the main text. A detailed discussion can be found in Appendix E.3.1. In the next section, we study the effect of regularization on SGD noise and fluctuation.

### 4.3 LEARNING WITH REGULARIZATION

Now, we show that regularization also causes a unique SGD noise. The loss function for $\Gamma - L_2$ regularized linear regression is

$$L_\Gamma(\mathbf{w}) = \frac{1}{2N}\sum_{i=1}^{N}\left[(\mathbf{w} - \mathbf{u})^{\text{T}}x_i\right]^2 + \frac{1}{2}\mathbf{w}^{\text{T}}\Gamma\mathbf{w} = \frac{1}{2}(\mathbf{w} - \mathbf{u})^{\text{T}}A(\mathbf{w} - \mathbf{u}) + \frac{1}{2}\mathbf{w}^{\text{T}}\Gamma\mathbf{w}, \quad (9)$$

where $\Gamma$ is a symmetric matrix; conventionally, one set $\Gamma = \gamma I_D$ with a scalar $\gamma > 0$. For conciseness, we assume that there is no noise in the label, namely $y_i = \mathbf{u}^{\text{T}}x_i$ with a constant vector $\mathbf{u}$. One important quantity in this case will be $\mathbf{u}\mathbf{u}^{\text{T}} := U$. The noise for this form of regularization can be calculated but takes a complicated form.

**Proposition 3.** (*Noise covariance matrix for learning with $L_2$ regularization*) *Let the algorithm be updated according to Eq. (1) on loss function (9) with $N \to \infty$ and $[A, \Gamma] = 0$. Then,*

$$C = \frac{1}{S}\left(A\Sigma A + \text{Tr}[A\Sigma]A + \text{Tr}[\Gamma'^{\text{T}}A\Gamma'U]A + \Gamma A'UA'\Gamma\right), \quad (10)$$

*where $A' := K^{-1}A$, $\Gamma' := K^{-1}\Gamma$ with $K := A + \Gamma$.*

Notice that the last term $\Gamma A'UA'\Gamma$ in $C$ is unique to the regularization-based noise: it is rank-1 because $U$ is rank-1. This term is due to the mismatch between the regularization and the minimum of the original loss. Also, note that the term $\text{Tr}[A\Sigma]$ is proportional to the training loss. Define the test loss to be $L_{\text{test}} := \lim_{t\to\infty} \mathbb{E}_{\mathbf{w}_t}\left[\frac{1}{2}(\mathbf{w}_t - \mathbf{u})^{\text{T}}A(\mathbf{w}_t - \mathbf{u})\right]$, we can prove the following theorem. We will show that one intriguing feature of discrete-time SGD is that the weight decay can be negative.

**Theorem 4.** (*Test loss and model fluctuation for $L_2$ regularization*) *Let the assumptions be the same as in Proposition 3. Then*

$$L_{\text{test}} = \frac{\lambda}{2S}\left(\text{Tr}[AK^{-2}\Gamma^2U]\kappa + r\right) + \frac{1}{2}\text{Tr}[AK^{-2}\Gamma^2U], \quad (11)$$

*where $\kappa := \frac{\text{Tr}[A^2K^{-1}G^{-1}]}{1 - \frac{\lambda}{S}\text{Tr}[A^2K^{-1}G^{-1}]}$, $r := \frac{\text{Tr}[A^3K^{-3}\Gamma^2G^{-1}U]}{1 - \frac{\lambda}{S}\text{Tr}[A^2K^{-1}G^{-1}]}$, with $G := 2I_D - \lambda\left(K + \frac{1}{S}K^{-1}A^2\right)$. Moreover, let $[\Gamma, U] = 0$, then*

$$\Sigma = \frac{\lambda}{S}\text{Tr}[AK^{-2}\Gamma^2U]\left(1 + \frac{\lambda\kappa}{S}\right)AK^{-1}G^{-1} + \frac{\lambda}{S}\left(A^2K^{-2}\Gamma^2U + \frac{\lambda r}{S}A\right)K^{-1}G^{-1}. \quad (12)$$

This result is numerically validated in Appendix A. The test loss (11) has an interesting consequence. One can show that there exist situations where the optimal $\Gamma$ is *negative*.[2] When discussing the test loss, we make the convention that if $\mathbf{w}_t$ diverges, then $L_{\text{test}} = \infty$.

**Corollary 1.** *Let $\gamma^* = \arg\min_\gamma L_{\text{test}}$. There exist $a$, $\lambda$ and $S$ such that $\gamma^* < 0$.*

The proof shows that when the learning rate is sufficiently large, only negative weight decay is allowed. This agrees with the argument in Liu et al. (2021) that discrete-time SGD introduces an implicit $L_2$ regularization that favors small norm solutions. A too-large learning rate requires a negative weight decay because a large learning rate already over-regularizes the model and one needs

---

[2]Some readers might argue that discussing test loss is meaningless when $N \to \infty$; however, this criticism does not apply because the size of the training set is not the only factor that affects generalization. In fact, this section's crucial message is that using a large learning rate affects the generalization by implicitly regularizing the model and, if one over-regularizes, one needs to offset this effect.

to introduce an explicit negative weight decay to offset this over-regularization effect of SGD. This is a piece of direct evidence that using a large learning rate can help regularize the models. It has been hypothesized that the dynamics of SGD implicitly regularizes neural networks such that the training favors simpler solutions (Kalimeris et al., 2019). Our result suggests one new mechanism for such a regularization.

## 5  NOISE STRUCTURE FOR GENERIC SETTINGS

The results in the previous sections suggest that (1) the SGD noises differ for different kinds of situations, and (2) SGD noise contains a term proportional to the training loss in general. These two facts motivate us to derive the noise covariance differently for different kinds of minima. Let $f(\mathbf{w}, x)$ denote the output of the model for a given input $x \in \mathbb{R}^D$. Here, we consider a more general case; $f(\mathbf{w}, x)$ may be any differentiable function, e.g., a non-linear deep neural network. The number of parameters in the model is denoted by $P$, and hence $\mathbf{w} \in \mathbb{R}^P$. For a training dataset $\{x_i, y_i\}_{i=1,2,...,N}$, the loss function with a $L_2$ regularization is given by

$$L_\Gamma(\mathbf{w}) = L_0(\mathbf{w}) + \frac{1}{2}\mathbf{w}^{\mathrm{T}}\Gamma\mathbf{w}, \tag{13}$$

where $L_0(\mathbf{w}) = \frac{1}{N}\sum_{i=1}^{N}\ell(f(\mathbf{w}, x_i), y_i)$ is the loss function without regularization, and $H_0$ is the Hessian of $L_0$. We focus on the MSE loss $\ell(f(\mathbf{w}, x_i), y_i) = [f(\mathbf{w}, x_i) - y_i]^2/2$. Our result crucially relies on the following two assumptions, which relate to the conditions of different kinds of local minima.

**Assumption 1.** (Fluctuation decays with batch size) $\Sigma$ is proportional to $S^{-1}$, i.e. $\Sigma = O(S^{-1})$.

This is justified by the results in all the related works (Liu et al., 2021; Xie et al., 2021; Meng et al., 2020; Mori et al., 2021), where $\Sigma$ is found to be $O(S^{-1})$.

**Assumption 2.** (Weak homogeneity) $|L - \ell_i|$ is small; in particular, it is of order $o(L)$.

This assumption amounts to assuming that the current training loss $L$ reflects the actual level of approximation for each data point well. In fact, since $L \geq 0$, one can easily show that $|L - \ell_i| = O(L)$. Here, we require a slightly stronger condition for a more clean expression, when $|L - \ell_i| = O(L)$ we can still get a similar expression but with some constant that hinders the clarity. Relaxing this condition can be an important and interesting future work. The above two conditions allow us to state our general theorem formally.

**Theorem 5.** *Let the training loss be $L_\Gamma = L_0 + \frac{1}{2}\mathbf{w}^{\mathrm{T}}\Gamma\mathbf{w}$ and the models be optimized with SGD in the neighborhood of a local minimum $\mathbf{w}^*$. Then,*

$$C(\mathbf{w}) = \frac{2L_0(\mathbf{w})}{S}H_0(\mathbf{w}) - \frac{1}{S}\nabla L_\Gamma(\mathbf{w})\nabla L_\Gamma(\mathbf{w})^{\mathrm{T}} + o(L_0). \tag{14}$$

The noise takes different forms for different kinds of local minima.

**Corollary 2.** *Omitting the terms of order $o(L_0)$, when $\Gamma \neq 0$,*

$$C = \frac{2L_0(\mathbf{w}^*)}{S}H_0(\mathbf{w}^*) - \frac{1}{S}\Gamma\mathbf{w}^*\mathbf{w}^{*\mathrm{T}}\Gamma + O(S^{-2}) + O(|\mathbf{w} - \mathbf{w}^*|^2). \tag{15}$$

*When $\Gamma = 0$ and $L_0(\mathbf{w}^*) \neq 0$,*

$$C = \frac{2L_0(\mathbf{w}^*)}{S}H_0(\mathbf{w}^*) + O(S^{-2}) + O(|\mathbf{w} - \mathbf{w}^*|^2). \tag{16}$$

*When $\Gamma = 0$ and $L_0(\mathbf{w}^*) = 0$,*

$$C = \frac{1}{S}\left(\mathrm{Tr}[H_0(\mathbf{w}^*)\Sigma]I_D - H_0(\mathbf{w}^*)\Sigma\right)H_0(\mathbf{w}^*) + O(S^{-2}) + O(|\mathbf{w} - \mathbf{w}^*|^2). \tag{17}$$

**Remark.** *Assumption 2 can be replaced by a weaker but more technical assumption called the "decoupling assumption", which has been used in recent works to derive the continuous-time distribution of SGD (Mori et al., 2021; Wojtowytsch, 2021). The Hessian approximation was invoked in most of the literature without considering the conditions of its applicability (Jastrzebski et al., 2018; Zhu et al., 2019; Liu et al., 2021; Wu et al., 2020; Xie et al., 2021). Our result does provide such*

*conditions for applicability. As indicated by the two assumptions, this theorem is applicable when the batch size is not too small and when the local minimum has a loss close to $0$. The reason for the failure of the Hessian approximation is that, while the FIM is equal to the expected Hessian $J = \mathbb{E}[H]$, there is no reason to expect the expected Hessian to be close to the actual Hessian of the minimum.*

The proof is given in Appendix C. Two crucial messages this corollary delivers are (1) the SGD noise is different in strength and shape in different kinds of local minima and that they need to be analyzed differently; (2) the SGD noise contains a term that is proportional to the training loss $L_0$ in general. Recently, it has been experimentally demonstrated that the SGD noise is indeed proportional to the training loss in realistic deep neural network settings, both when the loss function is MSE and cross-entropy (Mori et al., 2021); our result offers a theoretical justification. The previous works all treat all the minima as if the noise is similar (Jastrzebski et al., 2018; Zhu et al., 2019; Liu et al., 2021; Wu et al., 2020; Xie et al., 2021), which can lead to inaccurate or even incorrect understanding. For example, Theorem 3.2 in Xie et al. (2021) predicts a high escape probability from a sharp local or global minimum. However, this is incorrect because a model at a global minimum has zero probability of escaping due to a vanishing gradient. In contrast, the escape rate results derived in Mori et al. (2021) correctly differentiate the local and global minima. We also note that these general formulae are consistent with the exact solutions we obtained in the previous section than the Hessian approximation. For example, the dependence of the noise strength on the training loss in Theorem 2, and the rank-1 noise of regularization are all reflected in these formulae. In contrast, the simple Hessian approximation misses these crucial distinctions. Lastly, combining Theorem 5 with Theorem 1, one can also find the fluctuation.

**Corollary 3.** *Let the noise be as in Theorem 5, and omit the terms of order $O(S^{-2})$ and $O(|\mathbf{w} - \mathbf{w}^*|^2)$. Then, when $\Gamma \neq 0$ and when $\Lambda$, $H_0(\mathbf{w}^*)$ and $\Gamma$ commute with each other, $P_{r'}\Sigma = \frac{1}{S}\frac{\Lambda}{1-\mu}(2L_0 H_0 - \Gamma \mathbf{w}^* \mathbf{w}^{*\mathrm{T}}\Gamma)(H_0 + \Gamma)^+ \left[2I_D - \frac{\Lambda}{1+\mu}(H_0 + \Gamma)\right]^{-1}$. When $\Gamma = 0$ and $L_0(\mathbf{w}^*) \neq 0$, $P_r\Sigma = \frac{2L_0}{S(1-\mu)}P_r\Lambda\left(2I_D - \frac{\Lambda}{1+\mu}H_0\right)^{-1}$. When $\Gamma = 0$ and $L_0(\mathbf{w}^*) = 0$, $P_r\Sigma = 0$. Here the superscript $+$ is the Moore-Penrose pseudo inverse, $P_r := \mathrm{diag}(1, \ldots, 1, 0, \ldots, 0)$ is the projection operator with $r$ non-zero entries, $r \leq D$ is the rank of the Hessian $H_0$, and $r' \leq D$ is the rank of $H_0 + \Gamma$. For the null space $H_0$, $\Sigma$ can be arbitrary.*

## 6 Applications

One major advantage of analytical solutions is that they can be applied in a simple "plug-in" manner by the practitioners or theorists to analyze new problems they encounter. In this section, we briefly outline a few examples where the proposed theories can be relevant.

### 6.1 High-Dimensional Regression

We first apply our result to the high-dimensional regression problem and show how over-and-underparametrization might play a role in determining the minibatch noise. Here, we take $N, D \to \infty$ with the ratio $\alpha := N/D$ held fixed. The loss function is $L(\mathbf{w}) = \frac{1}{2N}\sum_{i=1}^N \left(\mathbf{w}^{\mathrm{T}}x_i - y_i\right)^2$. As in the standard literature (Hastie et al., 2019), we assume the existence of label noise: $y_i = \mathbf{u}^{\mathrm{T}}x_i + \epsilon_i$, with $\mathrm{Var}[\epsilon_i] = \sigma^2$. A key difference between our setting and the standard high-dimensional setting is that, in the standard setting (Hastie et al., 2019), one uses the GD algorithm with vanishing learning rate $\lambda$ instead of the minibatch SGD algorithm with a non-vanishing learning rate. Tackling the high-dimensional regression problem with non-vanishing $\lambda$ and a minibatch noise is another main technical contribution of this work. In this setting, we can obtain the following result on the noise covariance matrix.

**Proposition 4.** *Let $\hat{A} = \frac{1}{N}\sum_i^N x_i x_i^{\mathrm{T}}$ and suppose assumptions 1 and 2 hold. With fixed $S$, $\lambda$, then $C = \frac{1}{S}\left(\mathrm{Tr}[\hat{A}\Sigma]I_D - \hat{A}\Sigma\right)\hat{A} + \max\left\{0, \frac{\sigma^2}{S}\left(1 - \frac{1}{\alpha}\right)\right\}\hat{A}$.*

We note that this proposition follows from Theorem 5, showing an important theoretical application of our general theory. An interesting observation is that one $\Sigma$-independent term proportional to $\sigma^2$ emerges in the underparametrized regime ($\alpha > 1$). However, for the overparametrized regime, the noise is completely dependent on $\Sigma$, which is a sign that the stationary solution has no fluctuation. This shows that the degree of underparametrization also plays a distinctive role in the fluctuation. In fact, one can prove the following theorem, which is verified in Appendix A.2.

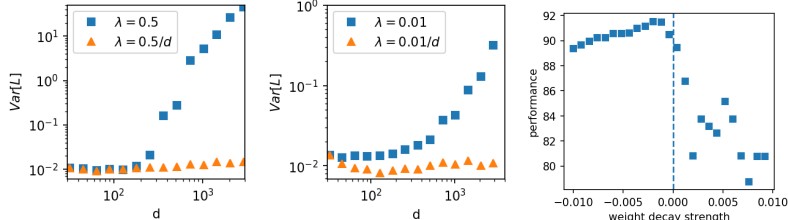

Figure 1: Realistic learning settings with neural networks and logistic regression. **Left**: Variance of training loss of a neural network with width $d$ and tanh activation on the MNIST dataset. We see that the variance explodes after $d \geq 200$. In contrast, rescaling the learning rate by $1/d$ results in a constant noise level in training. This suggests that the stability condition we derived for high-dimension regression is also useful for understanding deep learning. **Middle**: Stability of Adam with the same setting. Adam also experiences a similar stability problem when the model width increases. **Right**: Logistic regression on MNIST trained with SGD; with $\lambda = 1.5$, $S = 32$. We see that the optimal performance is also achieved at negative weight decay strength $\gamma$, suggesting that a large learning rate can indeed introduce effective regularization.

**Theorem 6.** *When a stationary solution exists for* **w**, *we have* $\text{Tr}[\hat{A}\Sigma] = \max\left\{0, \frac{\lambda\sigma^2}{S}\left(1 - \frac{1}{\alpha}\right)\hat{\kappa}\right\}$, *where* $\hat{\kappa} := \frac{\text{Tr}[\hat{G}^{-1}\hat{A}]}{1 - \frac{\lambda}{S}\text{Tr}[\hat{G}^{-1}\hat{A}]}$ *with* $\hat{G} := 2I_D - \lambda\left(1 - \frac{1}{S}\right)\hat{A}$.

### 6.2 IMPLICATION FOR NEURAL NETWORK TRAINING

It is commonly believed that the high-dimensional linear regression problem can be a minimal model for deep learning. Taking this stance, Theorem 6 suggests a technique for training neural networks. For SGD to converge, a positive semi-definite $\Sigma$ must exist; however, $\Sigma \geq 0$ if and only if $\hat{\kappa} \geq 0$. From $\hat{\kappa} > 0$, we have $\sum_{i=1}^{D} \frac{1}{2/\lambda a_i - 1 + 1/S} < S$, where $a_i$ are the eigenvalues of $\hat{A}$. This means that each summand should have the order of $D/S$. Thus the upper bound of $\lambda$ should have the order of $2S/aD$, where $a$ is the typical value of $a_i$'s. One implication of the dependence on the dimension is that the stability of a neural network trained with SGD may strongly depend on its width $d$, and one may rescale the learning rate according to the width to stabilize neural network training. See Figure 1-Left and Middle. We train a two-layer tanh neural network on MNIST and plot the variance of its training loss in the first epoch with fixed $\lambda = 0.5$. We see that, when $d \geq 200$, the training starts to destabilize, and the training loss begins to fluctuate dramatically. When rescaling the learning rate by $1/d$, we see that the variance of the training loss is successfully kept roughly constant across all $d$. This suggests a training technique worth being explored by practitioners in the field. In Figure 1-Middle, we also use Adam for training the same network and find a similar stabilizing trick to work for Adam.

### 6.3 A NATURAL LEARNING EXAMPLE WITH NEGATIVE WEIGHT DECAY

Sec. 4.3 shows that a too-large learning rate introduces an effective $L_2$ regularization that can be corrected by setting the weight decay to be negative. This effect can be observed in more realistic learning settings. We train a logistic regressor on the MNIST dataset with a large learning rate (of order $O(1)$). Figure 1-Right confirms that, at a large learning rate, the optimal weight decay can indeed be negative. This agrees with our argument that using a large learning rate can effectively regularize the training.

### 6.4 SECOND-ORDER METHODS

Understanding stochastic second-order methods (including the adaptive gradient methods) is also important for deep learning (Agarwal et al., 2017; Zhang and Liu, 2021; Martens, 2014; Kunstner et al., 2019). In this section, we apply our theory to two standard second-order methods: damped Newton's method (DNM) and natural gradient descent (NGD). We provide more accurate results than those derived in Liu et al. (2021). The derivations are given in Appendix D.2. For DNM, the preconditioning learning rate matrix is defined as $\Lambda := \lambda A^{-1}$. The model fluctuation is shown to be proportional to the inverse of the Hessian: $\Sigma = \frac{\lambda\sigma^2}{gS - \lambda D}A^{-1}$, where $g := 2(1 - \mu) - \left(\frac{1-\mu}{1+\mu} + \frac{1}{S}\right)\lambda$. The main difference with the previous results is that the fluctuation now depends explicitly on the dimension $D$, and implies a stability condition: $S \geq \lambda D/g$, corroborating the stability condition we derived above. For NGD, the preconditioning matrix is defined by the inverse of the Fisher information that $\Lambda := \frac{\lambda}{S}J(\mathbf{w})^{-1} = \frac{\lambda}{S}C^{-1}$. We show that $\Sigma = \frac{\lambda}{2}\left(\frac{1}{1+D}\frac{1}{1+\mu} + \frac{1}{1-\mu}\frac{1}{S}\right)A^{-1}$ is one solution when $\sigma = 0$, which also contains a correction related to $D$ compared to the result in Liu et al. (2021) which is $\Sigma = \frac{\lambda}{2}\left(\frac{1}{1+\mu} + \frac{1}{1-\mu}\frac{1}{S}\right)A^{-1}$. A consequence is that $J \sim \Sigma^{-1}$. The surprising

fact is that the stability of both NGD and DNM now crucially depends on $D$; combining with the results in Sec. 6.1, this suggests that the dimension of the problem may crucially affect the stability and performance of the minibatch-based algorithms. This result also implies that some features we derived are shared across many algorithms that depend on minibatch noise and that our results may be relevant to a broad class of optimization algorithms other than SGD.

### 6.5 FAILURE OF THE $\lambda - S$ SCALING LAW

One well-known technique in deep learning training is that one can scale $\lambda$ linearly as one increases the batch size $S$ to achieve high-efficiency training without hindering the generalization performance; however, it is known that this scaling law fails when the learning rate is too large, or the batch size is too small (Goyal et al., 2017). In Hoffer et al. (2017), this scaling law is established on the ground that $\Sigma \sim \lambda/S$. However, our result in Theorem 2 suggests the reason for the failure even for the simple setting of linear regression. Recall that the exact $\Sigma$ takes the form:

$$\Sigma = \frac{\lambda\sigma^2}{S}\left(1 + \frac{\kappa_\mu}{S}\right)G_\mu^{-1}$$

for a scalar $\lambda$. One notices that the leading term is indeed proportional to $\lambda/S$. However, the discrete-time SGD results in a second-order correction in $S$, and the term proportional to $1/S^2$ does not contain a corresponding $\lambda$; this explains the failure of the scaling law in small $S$, where the second-order contribution of $S$ becomes significant. To understand the failure at large $\lambda$, we need to look at the term $G_\mu$:

$$G_\mu = 2(1-\mu)I_D - \left(\lambda\frac{1-\mu}{1+\mu} + \frac{\lambda}{S}\right)A.$$

One notices that the second term contains a part that only depends on $\lambda$ but not on $S$. This part is negligible compared to the first term when $\lambda$ is small; however, it becomes significant as the second term approaches the first term. Therefore, increasing $\lambda$ changes this part of the fluctuation, and the scaling law no more holds if $\lambda$ is large.

### 6.6 POWER LAW TAIL IN DISCRETE-TIME SGD

It has recently been discovered that the SGD noise causes a heavy-tail distribution (Simsekli et al., 2019; 2020), with a tail decaying like a power law with tail index $\beta$ (Hodgkinson and Mahoney, 2020). In continuous-time, the stationary distribution has been found to obey a Student's t-like distribution, $p(w) \sim L^{-(1+\beta)/2} \sim \left(\sigma^2 + aw^2\right)^{-(1+\beta)/2}$ (Meng et al., 2020; Mori et al., 2021; Wojtowytsch, 2021). However, this result is only established for continuous-time approximations to SGD and one does not know what affects the exponent $\beta$ for discrete-time SGD. Our result in Theorem 2 can serve as a tool to find the discrete-time correction to the tail index of the stationary distribution. In Appendix D.3, we show that the tail index of discrete-time SGD in 1d can be estimated as $\beta(\lambda, S) = \frac{2S}{a\lambda} - S$. A clear discrete-time contribution is $-(S+1)$

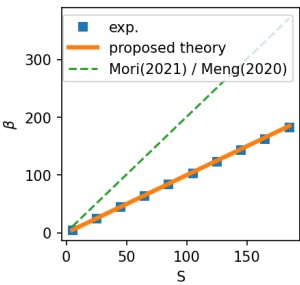

Figure 2: Comparison of the proposed theory with the continuous-time theory on the SGD stationary distribution for $a\lambda = 1$. The proposed theory agrees with the experiment exactly.

which depends only on the batch size, while $\frac{2S}{a\lambda} + 1$ is the tail index in the continuous-time limit (Mori et al., 2021). See Figure 2; the proposed formula agrees with the experiment. Knowing the tail index $\beta$ is important for understanding the SGD dynamics because $\beta$ is equal to the smallest moment of $w$ that diverges. For example, when $\beta \leq 4$, then the kurtosis of $w$ diverges, and one expects to see outliers of $w$ very often during training; when $\beta \leq 2$, then the second moment of $w$ diverges, and one does not expect $w$ to converge in the minimum under consideration. Our result suggests that the discrete-time dynamics always leads to a heavier tail than the continuous-time theory expects, and therefore is more unstable.

## 7 OUTLOOK

In this work, we have presented a systematic analysis with a focus on exactly solvable results to promote our fundamental understanding of SGD. One major limitation is that we have only focused on studying the asymptotic behavior of SGD in local minimum. For example, Ziyin et al. (2022) showed that SGD can converge to a local maximum when the learning rate is large. One important future step is thus to understand the SGD noise beyond a strongly convex landscape.

ACKNOWLEDGEMENT

Liu Ziyin thanks Jie Zhang, Junxia Wang, and Shoki Sugimoto. Ziyin is supported by the GSS Scholarship of The University of Tokyo. Kangqiao Liu was supported by the GSGC program of the University of Tokyo. This work was supported by KAKENHI Grant Numbers JP18H01145 and JP21H05185 from the Japan Society for the Promotion of Science.

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

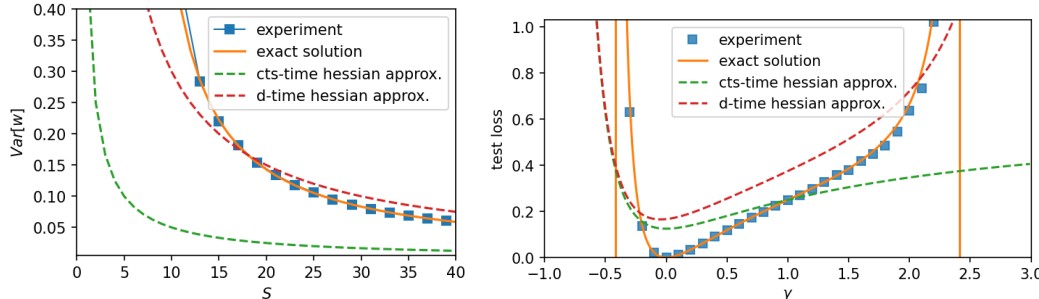

Figure 3: **Left**: 1d experiments with label noise. The parameters are set to be $a = 1.5$ and $\lambda = 1$. **Right**: Experiments with $L_2$ regularization with weight decay strength $\gamma$. The parameters are set to be $a = 1$, $\lambda = 0.5$, $S = 1$. This is the standard case with a vanishing optimal $\gamma$. The vertical lines show where our theory predicts a divergence.

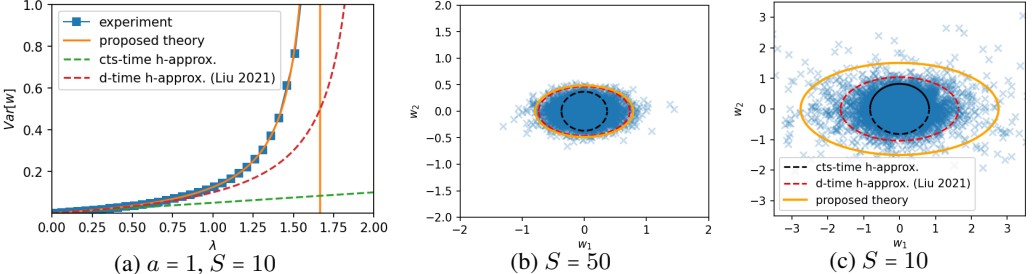

Figure 4: Comparison between theoretical predictions and experiments. (a) 1d experiment. We plot $\Sigma$ as an increasing function of $\lambda$. We see that the continuous-time approximation fails to predict the divergence at a learning rate and the prediction in Liu et al. (2021) severely underestimates the model fluctuation. In contrast, our result is accurate throughout the entire range of learning rates. (b)-(c) 2d experiments. The straight line shows where the proposed theory predicts a divergence in the variance, which agrees with experiment exactly. The Hessian has eigenvalues 1 and $0.5$, and $\lambda = 1.5$. For a large batch size, the discrete-time Hessian approximation is quite accurate; for a small $S$, the Hessian approximation underestimates the overall strength of the fluctuation. In contrast, the continuous-time result is both inaccurate in shape and in strength.

## A EXPERIMENTS

### A.1 LABEL NOISE AND REGULARIZATION

Theorem 2 can be verified empirically. We run 1d experiment in Figure 4(a) and high dimensional experiments in Figures 4(b)-(c), where we choose $D = 2$ for visualization. We see that the continuous Hessian approximation fails badly for both large and small batch sizes. When the batch size is large, both the discrete-time Hessian approximation and our solution give a accurate estimate of the shape and the spread of the distribution. This suggests that when the batch size is large, discreteness is the determining factor of the fluctuation. When the batch size is small, the discrete Hessian approximation severely underestimates the strength of the noise. This reflects the fact that the isotropic noise enhancement is dominant at a small batch size.

In Figure 3-Left, we run a 1d experiment with $\lambda = 1$, $N = 10000$ and $\sigma^2 = 0.25$. Comparing the predicted $\Sigma$, we see that the proposed theory agrees with the experiment across all ranges of $S$. The continuous theory with the Hessian approximation fails almost everywhere, while the recently proposed discrete theory with the Hessian approximation underestimates the fluctuation when $S$ is small. In Figure 3-Right, we plot a standard case where the optimal regularization strength $\gamma$ is vanishing.

Now, we validate the existence of the optimal negative weight decay as predicted by our formula. For illustration, we plot in Figure 5 the test loss (11) for a 1d example while varying either $S$ or $\lambda$. The orange vertical lines show the place where the theory predicts a divergence in the test loss. We also

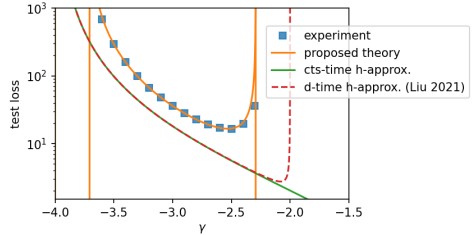

Figure 5: 1d experiments with $L_2$ regularization with weight decay strength $\gamma$. The parameters are set to be $a = 4$, $\lambda = 1$, $S = 64$. This shows a case where the optimal $\gamma$ is negative. The vertical lines show where our theory predicts a divergence.

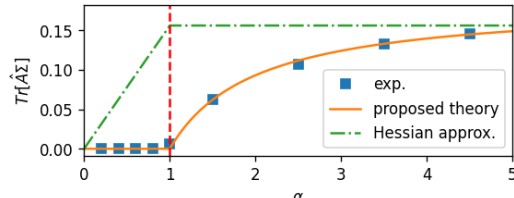

Figure 6: High-dimensional linear regression. We see that the predicted fluctuation coefficient agrees with the experiment well. The slight deviation is due to a finite training time and finite $N$ and $D$. On the other hand, a naive Hessian approximation results in a qualitatively wrong result.

plot a standard case where the optimal $\gamma$ is close to 0 in Appendix A. Also, we note that the proposed theory agrees better with the experiment.

## A.2 HIGH-DIMENSIONAL REGRESSION

See Figure 6-Left. We vary $N$ with $D = 1000$ held fixed. We set $\lambda = 0.01$ and $S = 32$. We see that the agreement between the theory and experiment is good, even for this modest dimension number $D$. The vertical line shows where the over-to-underparametrization transition takes place. As expected, there is no fluctuation when $\alpha < 1$, and the fluctuation gradually increases as $\alpha \to \infty$. On the other hand, the Hessian approximation gives a wrong picture, predicting fluctuation to rise when there is no fluctuation and predicting a constant fluctuation just when the fluctuation starts to rise.

Table 2: Comparison with previous results. For notational conciseness, we compare the case when all the relevant matrices commute. The model fluctuation $\Sigma$, the expected training loss $L_{\text{train}}$ and the expected test loss $L_{\text{test}}$ calculated by continuous- and discrete-time theories with Hessian approximation $C \approx H$ are presented. Exact solutions to these quantities obtained in the present work are shown in the rightmost column.

| | Hessian Approximation | | Exact Solution |
|---|---|---|---|
| | Cts-time Approximation | D-time Solution | This Work |
| | $\Sigma$ | $\Sigma$ | $\Sigma$ |
| Label Noise | $\frac{\lambda}{2S}I_D$ | $\frac{\lambda}{S}(2I_D - \lambda A)^{-1}$ | $\frac{\lambda\sigma^2}{S}\left(1 + \frac{\lambda\kappa}{S}\right)\left[2I_D - \lambda\left(1 + \frac{1}{S}\right)A\right]^{-1}$ |
| Input Noise | $\frac{\lambda}{2S}I_D$ | $\frac{\lambda}{S}(2I_D - \lambda K)^{-1}$ | $\frac{\lambda\text{Tr}[AK^{-1}BU]}{S}\left(1 + \frac{\lambda\kappa'}{S}\right)\left[2I_D - \lambda\left(1 + \frac{1}{S}\right)K\right]^{-1}$ |
| $L_2$ Regularization | $\frac{\lambda}{2S}I_D$ | $\frac{\lambda}{S}(2I_D - \lambda K)^{-1}$ | Eq. (12) |
| | $L_{\text{train}}$ | $L_{\text{train}}$ | $L_{\text{train}}$ |
| Label Noise | $\frac{\lambda}{4S}\text{Tr}[A] + \frac{1}{2}\sigma^2$ | Eq. (20) | $\frac{\sigma^2}{2}\left(1 + \frac{\lambda\kappa}{S}\right)$ |
| Input Noise | $\frac{\lambda}{4S}\text{Tr}[K] + \frac{1}{2}\text{Tr}[AK^{-1}BU]$ | Eq. (28) | $\frac{1}{2}\text{Tr}[AK^{-1}BU]\left(1 + \frac{\lambda}{S}\kappa'\right)$ |
| $L_2$ Regularization | $\frac{\lambda}{4S}\text{Tr}[K] + \frac{1}{2}\text{Tr}[AK^{-1}\Gamma U]$ | Eq. (37) | Eq. (151) |
| | $L_{\text{test}}$ | $L_{\text{test}}$ | $L_{\text{test}}$ |
| Label Noise | $\frac{\lambda}{4S}\text{Tr}[A]$ | $\frac{\lambda}{2S}\text{Tr}[A(2I_D - \lambda A)^{-1}]$ | $\frac{\lambda\sigma^2}{2S}\kappa$ |
| Input Noise | $\frac{\lambda}{4S}\text{Tr}[A] + \frac{1}{2}\text{Tr}[B'^{\text{T}}AB'U]$ | Eq. (29) | $\frac{\lambda}{2S}\text{Tr}[AK^{-1}BU]\kappa' + \frac{1}{2}\text{Tr}[B'^{\text{T}}AB'U]$ |
| $L_2$ Regularization | $\frac{\lambda}{4S}\text{Tr}[A] + \frac{1}{2}\text{Tr}[AK^{-2}\Gamma^2 U]$ | Eq. (38) | Eq. (11) |

## B    COMPARISON WITH CONVENTIONAL HESSIAN APPROXIMATION

We compare our results for the three cases with the results obtained with the conventional Hessian approximation of the noise covariance, i.e., $C \approx H$, where $H$ is the Hessian of the loss function. We summarize the analytical results for a special case in Table 2.

### B.1    LABEL NOISE

We first consider discrete-time dynamics with the Hessian approximation. The matrix equation is

$$\Sigma A + A\Sigma - \lambda A\Sigma A = \frac{\lambda}{S}A. \tag{18}$$

Compared with the exact result (3), it is a large-$S$ limit up to the constant $\sigma^2$. This constant factor is ignored during the approximation that $J(\mathbf{w}) := \mathbb{E}_{\text{B}}[\nabla l \nabla l^{\text{T}}] \approx \mathbb{E}_{\text{B}}[\nabla \nabla^{\text{T}} l] := H(\mathbf{w})$, which is exact only when $l(\{x_i\}, \mathbf{w})$ is a *negative log likelihood* function of $\mathbf{w}$. Solving the matrix equation yields

$$\Sigma = \frac{\lambda}{S}(2I_D - \lambda A)^{-1}. \tag{19}$$

The training loss and the test loss are

$$L_{\text{train}} = \frac{\lambda}{2S}\text{Tr}[A(2I_D - \lambda A)^{-1}] + \frac{1}{2}\sigma^2, \tag{20}$$

$$L_{\text{test}} = \frac{\lambda}{2S}\text{Tr}[A(2I_D - \lambda A)^{-1}]. \tag{21}$$

On the other hand, by taking the large-$S$ limit directly from the exact equation (3), the factor $\sigma^2$ is present:

$$\Sigma A + A\Sigma - \lambda A\Sigma A = \frac{\lambda}{S}\sigma^2 A. \tag{22}$$

For the continuous-time limit with the Hessian approximation, the matrix equation is

$$\Sigma A + A\Sigma = \frac{\lambda}{S}A, \tag{23}$$

which is the small-$\lambda$ limit up to the factor $\sigma^2$. The variance is

$$\Sigma = \frac{\lambda}{2S}I_D. \tag{24}$$

The training and the test error are

$$L_{\text{train}} = \frac{\lambda}{4S}\text{Tr}[A] + \frac{1}{2}\sigma^2, \tag{25}$$

$$L_{\text{test}} = \frac{\lambda}{4S}\text{Tr}[A]. \tag{26}$$

Again, taking the small-$\lambda$ limit directly from the exact result (3) shows the presence of the factor $\sigma^2$ on the right hand side of the matrix equation.

## B.2    INPUT NOISE

The case with the input noise is similar to the label noise. This can be understood if we replace $A$ by $K$ and $\sigma^2$ by $\text{Tr}[AK^{-1}BU]$. The model parameter variance resulting from the discrete-time dynamics under the Hessian approximation is

$$\Sigma = \frac{\lambda}{S}(2I_D - \lambda K)^{-1}. \tag{27}$$

The training and the test error are

$$L_{\text{train}} = \frac{\lambda}{2S}\text{Tr}[K(2I_D - \lambda K)^{-1}] + \frac{1}{2}\text{Tr}[AK^{-1}BU], \tag{28}$$

$$L_{\text{test}} = \frac{\lambda}{2S}\text{Tr}[A(2I_D - \lambda K)^{-1}] + \frac{1}{2}\text{Tr}[B'^{\text{T}}AB'U]. \tag{29}$$

The large-$S$ limit from the exact matrix equation (144) results in a prefactor $\text{Tr}[AK^{-1}BU]$ in the fluctuation:

$$\Sigma = \frac{\lambda}{S}\text{Tr}[AK^{-1}BU](2I_D - \lambda K)^{-1}. \tag{30}$$

For the continuous-time limit, we take $\lambda \to 0$. The Hessian approximation gives

$$\Sigma = \frac{\lambda}{2S}I_D, \tag{31}$$

$$L_{\text{train}} = \frac{\lambda}{4S}\text{Tr}[K] + \frac{1}{2}\text{Tr}[AK^{-1}BU], \tag{32}$$

$$L_{\text{test}} = \frac{\lambda}{4S}\text{Tr}[A] + \frac{1}{2}\text{Tr}[B'^{\text{T}}AB'U], \tag{33}$$

The large-$S$ limit again produces a prefactor $\text{Tr}[AK^{-1}BU]$.

## B.3    $L_2$ REGULARIZATION

For learning with regularization, there is a more difference between the Hessian approximation and the limit taken directly from the exact theory. We first adopt the Hessian approximation for the discrete-time dynamics. The matrix equation is

$$\Sigma K + K\Sigma - \lambda K\Sigma K = \frac{\lambda}{S}K, \tag{34}$$

which is similar to the previous subsection. However, it is different from the large-$S$ limit of the exact matrix equation (154):

$$\Sigma K + K\Sigma - \lambda K\Sigma K = \frac{\lambda}{S}\left(\text{Tr}[AK^{-2}\Gamma^2 U]A + AK^{-1}\Gamma U\Gamma K^{-1}A\right). \tag{35}$$

This significant difference suggests that the conventional Fisher-to-Hessian approximation $J \approx H$ fails badly. The fluctuation, the training loss, and the test loss with the Hessian approximation are

$$\Sigma = \frac{\lambda}{S}(2I_D - \lambda K)^{-1}, \tag{36}$$

$$L_{\text{train}} = \frac{\lambda}{2S}\text{Tr}[K(2I_D - \lambda K)^{-1}] + \frac{1}{2}\text{Tr}[AK^{-1}\Gamma U], \tag{37}$$

$$L_{\text{test}} = \frac{\lambda}{2S}\text{Tr}[A(2I_D - \lambda K)^{-1}] + \frac{1}{2}\text{Tr}[AK^{-2}\Gamma^2 U], \tag{38}$$

while the large-$S$ limit of the exact theory yields

$$\Sigma = \frac{\lambda}{S}\mathrm{Tr}[AK^{-2}\Gamma^2 U]AK^{-1}(2I_D - \lambda K)^{-1} + \frac{\lambda}{S}A^2 K^{-3}\Gamma^2(2I_D - \lambda K)^{-1}U, \tag{39}$$

$$L_{\text{train}} = \frac{\lambda}{2S}\mathrm{Tr}[AK^{-2}\Gamma^2 U]\mathrm{Tr}[A(2I_D - \lambda K)^{-1}] + \frac{\lambda}{2S}\mathrm{Tr}[A^2 K^{-2}\Gamma^2(2I_D - \lambda K)^{-1}U]$$
$$+ \frac{1}{2}\mathrm{Tr}[AK^{-1}\Gamma U], \tag{40}$$

$$L_{\text{test}} = \frac{\lambda}{2S}\mathrm{Tr}[AK^{-2}\Gamma^2 U]\mathrm{Tr}[A(2I_D - \lambda K)^{-1}] + \frac{\lambda}{2S}\mathrm{Tr}[A^3 K^{-3}\Gamma^2(2I_D - \lambda K)^{-1}U]$$
$$+ \frac{1}{2}\mathrm{Tr}[AK^{-2}\Gamma^2 U]. \tag{41}$$

The continuous-time results are obtained by taking the small-$\lambda$ limit on Eqs. (36)-(38) for the Hessian approximation and on Eqs. (39)-(41) for the limiting cases of the exact theory. Specifically, for the Hessian approximation, we have

$$\Sigma = \frac{\lambda}{2S}I_D, \tag{42}$$

$$L_{\text{train}} = \frac{\lambda}{4S}\mathrm{Tr}[K] + \frac{1}{2}\mathrm{Tr}[AK^{-1}\Gamma U], \tag{43}$$

$$L_{\text{test}} = \frac{\lambda}{4S}\mathrm{Tr}[A] + \frac{1}{2}\mathrm{Tr}[AK^{-2}\Gamma^2 U]. \tag{44}$$

The small-$\lambda$ limit of the exact theory yields

$$\Sigma = \frac{\lambda}{2S}\mathrm{Tr}[AK^{-2}\Gamma^2 U]AK^{-1} + \frac{\lambda}{2S}A^2 K^{-3}\Gamma^2 U, \tag{45}$$

$$L_{\text{train}} = \frac{\lambda}{4S}\mathrm{Tr}[AK^{-2}\Gamma^2 U]\mathrm{Tr}[A] + \frac{\lambda}{4S}\mathrm{Tr}[A^2 K^{-2}\Gamma^2 U] + \frac{1}{2}\mathrm{Tr}[AK^{-1}\Gamma U], \tag{46}$$

$$L_{\text{test}} = \frac{\lambda}{4S}\mathrm{Tr}[AK^{-2}\Gamma^2 U]\mathrm{Tr}[A] + \frac{\lambda}{4S}\mathrm{Tr}[A^3 K^{-3}\Gamma^2 U] + \frac{1}{2}\mathrm{Tr}[AK^{-2}\Gamma^2 U]. \tag{47}$$

## C    PROOF OF THE GENERAL FORMULA

### C.1    PROOF OF THEOREM 5 AND COROLLARY 2

We restate the theorem.

**Theorem 7.** *Let the training loss be $L_\Gamma = L_0 + \frac{1}{2}\mathbf{w}^{\mathrm{T}}\Gamma\mathbf{w}$ and the models be optimized with SGD in the neighborhood of a local minimum $\mathbf{w}^*$. When $\Gamma \neq 0$, the noise covariance is given by*

$$C = \frac{2L_0(\mathbf{w}^*)}{S}H_0(\mathbf{w}^*) - \frac{1}{S}\Gamma\mathbf{w}^*\mathbf{w}^{*\mathrm{T}}\Gamma + O(S^{-2}) + O(|\mathbf{w} - \mathbf{w}^*|^2). \tag{48}$$

*When $\Gamma = 0$ and $L_0(\mathbf{w}^*) \neq 0$,*

$$C = \frac{2L_0(\mathbf{w}^*)}{S}H_0(\mathbf{w}^*) + O(S^{-2}) + O(|\mathbf{w} - \mathbf{w}^*|^2). \tag{49}$$

*When $\Gamma = 0$ and $L_0(\mathbf{w}^*) = 0$,*

$$C = \frac{1}{S}\left(\mathrm{Tr}[H_0(\mathbf{w}^*)\Sigma]I_D - H_0(\mathbf{w}^*)\Sigma\right)H_0(\mathbf{w}^*) + O(S^{-2}) + O(|\mathbf{w} - \mathbf{w}^*|^2). \tag{50}$$

*Proof.* We will use the following shorthand notations: $\ell_i := \ell(f(\mathbf{w}, x_i), y_i)$, $\ell_i' := \frac{\partial \ell_i}{\partial f}$, $\ell_i'' := \frac{\partial^2 \ell_i}{\partial f^2}$. The Hessian of the loss function without regularization $H_0(\mathbf{w}) = \nabla\nabla^{\mathrm{T}}L_0(\mathbf{w})$ is given by

$$H_0(\mathbf{w}) = \frac{1}{N}\sum_{i=1}^{N}\ell_i''\nabla f(\mathbf{w}, x_i)\nabla f(\mathbf{w}, x_i)^{\mathrm{T}} + \frac{1}{N}\sum_{i=1}^{N}\ell_i'\nabla\nabla^{\mathrm{T}}f(\mathbf{w}, x_i). \tag{51}$$

The last term of Eq. (51) can be ignored when $L_0 \ll 1$, since

$$\left\|\frac{1}{N}\sum_{i=1}^{N}\ell_i'\nabla\nabla^{\mathrm{T}}f(\mathbf{w}, x_i)\right\|_F \leq \left(\frac{1}{N}\sum_{i=1}^{N}(\ell_i')^2\right)^{1/2}\left(\frac{1}{N}\sum_{i=1}^{N}\|\nabla\nabla^{\mathrm{T}}f(\mathbf{w}, x_i)\|_F^2\right)^{1/2}$$

$$= \langle \ell'^2 \rangle^{1/2}\left(\frac{1}{N}\sum_{i=1}^{N}\|\nabla\nabla^{\mathrm{T}}f(\mathbf{w}, x_i)\|_F^2\right)^{\frac{1}{2}},$$

$$= \sqrt{2L_0(\mathbf{w})}\left(\frac{1}{N}\sum_{i=1}^{N}\|\nabla\nabla^{\mathrm{T}}f(\mathbf{w}, x_i)\|_F^2\right)^{\frac{1}{2}},$$

where $\|\cdot\|_F$ stands for the Frobenius norm[3], and we have defined the variable $\langle \ell'^2 \rangle := \frac{1}{N}\sum_{i=1}^{N}(\ell_i')^2$. Since $\ell_i'' = 1$ for the mean-square error, we obtain

$$H_0(\mathbf{w}) = \frac{1}{N}\sum_{i=1}^{N}\nabla f(\mathbf{w}, x_i)\nabla f(\mathbf{w}, x_i)^{\mathrm{T}} + O\left(\sqrt{L_0}\right) \tag{52}$$

near a minimum. The Hessian with regularization $H_\Gamma(\mathbf{w}) = \nabla\nabla^{\mathrm{T}}L_\Gamma(\mathbf{w})$ is just given by $H_0(\mathbf{w}) + \Gamma$.

On the other hand, the SGD noise covariance $C(\mathbf{w})$ is given by Eq. (2). By assumption 2, the SGD noise covariance is directly related to the Hessian:

$$C(\mathbf{w}) = \frac{\langle \ell'^2 \rangle}{SN}\sum_{i=1}^{N}\nabla f(\mathbf{w}, x_i)\nabla f(\mathbf{w}, x_i)^{\mathrm{T}} - \frac{1}{S}\nabla L_\Gamma(\mathbf{w})\nabla L_\Gamma(\mathbf{w})^{\mathrm{T}}$$

$$+ \frac{2}{SN}\sum_{i=1}^{N}(\ell_i - L_0)\nabla f(\mathbf{w}, x_i)\nabla f(\mathbf{w}, x_i)^{\mathrm{T}}$$

$$= \frac{2L_0(\mathbf{w})}{S}H_0(\mathbf{w}) - \frac{1}{S}\nabla L_\Gamma(\mathbf{w})\nabla L_\Gamma(\mathbf{w})^{\mathrm{T}} + o(L_0). \tag{53}$$

This finishes the proof. □

---

[3]In the linear regression problem, the last term of Eq. (51) does not exist since $\nabla\nabla^{\mathrm{T}}f(\mathbf{w}, x_i) = 0$.

Now we prove Corollary 2.

*Proof.* Near a minimum $\mathbf{w}^*$ of the full loss $L_\Gamma(\mathbf{w})$, we have

$$\nabla L_\Gamma(\mathbf{w}) = H_0(\mathbf{w}^*)(\mathbf{w} - \mathbf{w}^*) + \Gamma\mathbf{w}^* + O(|\mathbf{w} - \mathbf{w}^*|^2), \tag{54}$$

within the approximation $L_\Gamma(\mathbf{w}) = L_\Gamma(\mathbf{w}^*) + (1/2)(\mathbf{w} - \mathbf{w}^*)^{\mathrm{T}} H_\Gamma(\mathbf{w}^*)(\mathbf{w} - \mathbf{w}^*)^{\mathrm{T}} + O(|\mathbf{w} - \mathbf{w}^*|^2)$. Equations (14) and (54) give the SGD noise covariance near a minimum of $L_\Gamma(\mathbf{w})$.

Now it is worth discussing two different cases separately: (1) with regularization and (2) without regularization. We first discuss the case when regularization is present. In this case, the regularization $\Gamma$ is not small enough, and the SGD noise covariance is *not* proportional to the Hessian. Near a local or global minimum $\mathbf{w} \approx \mathbf{w}^*$, the first term of the right-hand side of Eq. (54) is negligible, and hence we obtain

$$\begin{aligned}
\mathbb{E}_{\mathbf{w}}[C(\mathbf{w})] &= \frac{2L_0(\mathbf{w}^*)}{S} H_0(\mathbf{w}^*) - \frac{1}{S}\Gamma\mathbf{w}^*\mathbf{w}^{*\mathrm{T}}\Gamma \\
&\quad + \mathbb{E}_{\mathbf{w}}\left[\frac{1}{S} H_0(\mathbf{w}^*)(\mathbf{w} - \mathbf{w}^*)(\mathbf{w} - \mathbf{w}^*)^{\mathrm{T}} H_0(\mathbf{w}^*)\right] + O(|\mathbf{w} - \mathbf{w}^*|^2) \\
&= \frac{2L_0(\mathbf{w}^*)}{S} H_0(\mathbf{w}^*) - \frac{1}{S}\Gamma\mathbf{w}^*\mathbf{w}^{*\mathrm{T}}\Gamma + O(S^{-2}) + O(|\mathbf{w} - \mathbf{w}^*|^2). 
\end{aligned} \tag{55}$$

where we have used the fact that $\mathbb{E}[\mathbf{w}] = \mathbf{w}^*$. The SGD noise does not vanish even at a global minimum of $L_\Gamma(\mathbf{w})$. Note that this also agrees with the exact result derived in Sec. 4.3: together with an anisotropic noise that is proportional to the Hessian, a rank-1 noise proportional to the strength of the regularization appears. This rank-1 noise is a signature of regularization.

On the other hand, as we will see below, the SGD noise covariance is proportional to the Hessian near a minimum when there is no regularization, i.e., $\Gamma = 0$. We have

$$C(\mathbf{w}) = \frac{2L_0(\mathbf{w})}{S} H_0(\mathbf{w}) - \frac{1}{S} H_0(\mathbf{w}^*)(\mathbf{w} - \mathbf{w}^*)(\mathbf{w} - \mathbf{w}^*)^{\mathrm{T}} H_0(\mathbf{w}^*) + O(|\mathbf{w} - \mathbf{w}^*|^2). \tag{56}$$

For this case, we need to differentiate between a local minimum and a global minimum. When $L_0(\mathbf{w}^*)$ is not small enough (e.g. at a local but not global minimum),

$$\begin{aligned}
C(\mathbf{w}) &= \frac{2L_0(\mathbf{w}^*)}{S} H_0(\mathbf{w}) + \frac{(\mathbf{w} - \mathbf{w}^*)^{\mathrm{T}} H_0(\mathbf{w}^*)(\mathbf{w} - \mathbf{w}^*)}{S} H_0(\mathbf{w}) \\
&\quad - \frac{1}{S} H_0(\mathbf{w}^*)(\mathbf{w} - \mathbf{w}^*)(\mathbf{w} - \mathbf{w}^*)^{\mathrm{T}} H_0(\mathbf{w}^*) + O(|\mathbf{w} - \mathbf{w}^*|^2) \\
&= \frac{2L_0(\mathbf{w}^*)}{S} H_0(\mathbf{w}) + O(S^{-2}) + O(|\mathbf{w} - \mathbf{w}^*|^2) \\
&= \frac{2L_0(\mathbf{w}^*)}{S} H_0(\mathbf{w}^*) + O(S^{-2}) + O(|\mathbf{w} - \mathbf{w}^*|^2), 
\end{aligned} \tag{57}$$

and so, to leading order,

$$C = \frac{2L_0(\mathbf{w}^*)}{S} H_0(\mathbf{w}^*), \tag{58}$$

which is proportional to the Hessian but also proportional to the achievable approximation error.

On the other hand, when $L_0(\mathbf{w}^*)$ is vanishingly small (e.g. at a global minimum), we have $2L_0(\mathbf{w}) \approx (\mathbf{w} - \mathbf{w}^*)^{\mathrm{T}} H_0(\mathbf{w}^*)(\mathbf{w} - \mathbf{w}^*)$, and thus obtain

$$\begin{aligned}
C(\mathbf{w}) &= \frac{1}{S}\left[(\mathbf{w} - \mathbf{w}^*)^{\mathrm{T}} H_0(\mathbf{w}^*)(\mathbf{w} - \mathbf{w}^*) H_0(\mathbf{w}^*) - H_0(\mathbf{w}^*)(\mathbf{w} - \mathbf{w}^*)(\mathbf{w} - \mathbf{w}^*)^{\mathrm{T}} H_0(\mathbf{w}^*)\right] \\
&\quad + O(S^{-2}) + O(|\mathbf{w} - \mathbf{w}^*|^2), 
\end{aligned} \tag{59}$$

i.e.,

$$\mathbb{E}[C] = \frac{1}{S}\left(\mathrm{Tr}[H_0\Sigma] I_D - H_0\Sigma\right) H_0 + O(S^{-2}) + O(|\mathbf{w} - \mathbf{w}^*|^2). \tag{60}$$

This completes the proof. $\square$

**Remark.** *It should be noted that the second term on the right-hand side of Eq. (59) would typically be much smaller than the first term for large $D$. For example, when $H_0(\mathbf{w}^*) = aI_D$ with $a > 0$, the first and the second terms are respectively given by $(a^2/S)\|\mathbf{w}-\mathbf{w}^*\|^2 I_D$ and $-(a^2/S)(\mathbf{w}-\mathbf{w}^*)(\mathbf{w}-\mathbf{w}^*)^\mathrm{T}$. The Frobenius norm of the former is given by $(Da^2/S)\|\mathbf{w} - \mathbf{w}^*\|^2$, while that of the latter is given by $(a^2/S)\|\mathbf{w} - \mathbf{w}^*\|^2$, which indicates that in Eq. (59), the first term is dominant over the second term for large $D$. Therefore the second term of Eq. (59) can be dropped for large $D$, and Eq. (59) is simplified as*

$$\begin{cases} C(\mathbf{w}) \approx \frac{(\mathbf{w}-\mathbf{w}^*)^\mathrm{T} H_0(\mathbf{w}^*)(\mathbf{w}-\mathbf{w}^*)}{S} H_0(\mathbf{w}^*); \\ \mathbb{E}[C] \approx \frac{\mathrm{Tr}[H_0\Sigma]}{S} H_0. \end{cases} \tag{61}$$

*Again, the SGD noise covariance is proportional to the Hessian.*

In conclusion, as long as the regularization is small enough, that the SGD noise covariance near a minimum is proportional to the Hessian is a good approximation. This implies that the noise is multiplicative, which is known to lead to a heavy tail distribution (Clauset et al., 2009; Levy and Solomon, 1996). Thus, we have studied the nature of the minibatch SGD noise in three different situations. As an example, we have demonstrated the power of this general formulation by applying it to the high-dimensional linear regression problem in Sec. 6.1.

## C.2  PROOF OF COROLLARY 3

*Proof.* We prove the case where $\Gamma = 0$ and $L(\mathbf{w}^*) \neq 0$ as an example. Substituting Theorem 5 into Theorem 1 yields

$$\left[ 2I_D - \frac{1}{1+\mu}\Lambda H_0 \right] \Lambda H_0 \Sigma = \frac{2L_0}{S(1-\mu)}\Lambda^2 H_0, \tag{62}$$

where we have assumed necessary commutation relations. Suppose that the Hessian $H_0$ is of rank-$r$ with $r \leq D$. The singular-value decomposition and its Moore-Penrose pseudo inverse are given by $H_0 = USV^\mathrm{T}$ and $H_0^+ = VS^+U^\mathrm{T}$, respectively, where $U$ and $V$ are unitary, $S$ is a rank-$r$ diagonal matrix with elements being singular values of $H_0$, and $S^+$ is obtained by inverting every non-zero entry of $S$. Multiplying $H_0^+$ to both sides of the above equation, we have

$$P_r\Sigma = \frac{2L_0}{S(1-\mu)}P_r\Lambda\left(2I_D - \frac{\Lambda}{1+\mu}H_0\right)^{-1}, \tag{63}$$

where $P_r = \mathrm{diag}(1,1,\ldots,1,0,\ldots,0)$ is the projection operator with $r$ non-zero entries. When the Hessian is full-rank, i.e., $r = D$, the Moore-Penrose pseudo inverse is nothing but the usual inverse. The other cases can be calculated similarly. □

# D APPLICATIONS

## D.1 INFINITE-DIMENSIONAL LIMIT OF THE LINEAR REGRESSION PROBLEM

Now we apply the general theory in Sec. 5 to linear regressions in the high-dimensional limit, namely $N, D \to \infty$ with $\alpha := N/D$ held fixed.

### D.1.1 PROOF OF PROPOSITION 4

The loss function

$$L(\mathbf{w}) = \frac{1}{2N} \sum_{i=1}^{N} \left( \mathbf{w}^{\mathrm{T}} x_i - y_i \right)^2 \tag{64}$$

with $y_i = \mathbf{u}^{\mathrm{T}} + \epsilon_i$ can be written as

$$L(\mathbf{w}) = \frac{1}{2} \left( \mathbf{w} - \mathbf{u} - \hat{A}^+ \mathbf{v} \right)^{\mathrm{T}} \hat{A} \left( \mathbf{w} - \mathbf{u} - \hat{A}^+ \mathbf{v} \right) - \frac{1}{2} \mathbf{v}^{\mathrm{T}} \hat{A}^+ \mathbf{v} + \frac{1}{2N} \sum_{i=1}^{N} \epsilon_i^2, \tag{65}$$

where $\hat{A} := \frac{1}{N} \sum_{i=1}^{N} x_i x_i^{\mathrm{T}}$ is an empirical covariance for the training data and $\mathbf{v} := \frac{1}{N} \sum_{i=1}^{N} x_i \epsilon_i$. The symbol $(\cdot)^+$ denotes the Moore-Penrose pseudoinverse. We also introduce the the averaged traing loss: $L_{\mathrm{train}} := \mathbb{E}_{\mathbf{w}}[L(\mathbf{w})]$

The minimum of the loss function is given by

$$\mathbf{w}^* = \mathbf{u} + \hat{A}^+ \mathbf{v} + \Pi \mathbf{r}, \tag{66}$$

where $\mathbf{r} \in \mathbb{R}^D$ is an arbitrary vector and $\Pi$ is the projection onto the null space of $\hat{A}$. Since $1 - \Pi = \hat{A}^+ \hat{A}$, $\mathbf{w}^*$ is also expressed as

$$\mathbf{w}^* = \hat{A}^+(\hat{A}\mathbf{u} + \mathbf{v}) + \Pi \mathbf{r}. \tag{67}$$

In an underparameterized regime $\alpha > 1$, $\Pi = 0$ almost surely holds as long as the minimum eigenvalue of $A$ (not $\hat{A}$) is positive (Hastie et al., 2019). In this case, $\hat{A}^+ = \hat{A}^{-1}$ and we obtain

$$\mathbf{w}^* = \mathbf{u} + \hat{A}^{-1} \mathbf{v} \quad \text{for } \alpha > 1. \tag{68}$$

On the other hand, in an overparameterized regime $\alpha > 1$, $\Pi \neq 0$ and there are infinitely many global minima. In the ridgeless regression, we consider the global minimum that has the minimum norm $\|\mathbf{w}^*\|$, which corresponds to

$$\mathbf{w}^* = \hat{A}^+(\hat{A}\mathbf{u} + \mathbf{v}) = (1 - \Pi)\mathbf{u} + \hat{A}^+ \mathbf{v} \quad \text{for ridgeless regression with } \alpha < 1. \tag{69}$$

In both cases, the loss function is expressed as

$$L(\mathbf{w}) = \frac{1}{2} \left( \mathbf{w} - \mathbf{w}^* \right)^{\mathrm{T}} \hat{A} \left( \mathbf{w} - \mathbf{w}^* \right) - \frac{1}{2} \mathbf{v}^{\mathrm{T}} \hat{A} \mathbf{v} + \frac{1}{2N} \sum_{i=1}^{N} \epsilon_i^2. \tag{70}$$

Asymptotically, $\mathbf{w}_t$ converges to a stationary point $\mathbf{w}^*$ with fluctuation $\Sigma$ obeying the following equation (Theorem 1:

$$\lambda \hat{A} \Sigma + \lambda \Sigma \hat{A} - \lambda^2 \hat{A} \Sigma \hat{A} = \lambda^2 C. \tag{71}$$

The SGD noise covariance $C$ is given by Eq. (14). In the present case, the Hessian is given by $H = \hat{A}$ and we also have

$$\frac{1}{N} \sum_{i=1}^{N} (\ell_i')^2 = \frac{1}{N} \sum_{i=1}^{N} \left( \mathbf{w}^{\mathrm{T}} x_i - y_i \right)^2 = \frac{2}{N} \sum_{i=1}^{N} \ell_i = 2L(\mathbf{w}). \tag{72}$$

On the other hand, $\nabla L(\mathbf{w}) \nabla L(\mathbf{w})^{\mathrm{T}} = \hat{A}(\mathbf{w} - \mathbf{w}^*)(\mathbf{w} - \mathbf{w}^*)^{\mathrm{T}} \hat{A}$, and hence $\mathbb{E}_{\mathbf{w}}[\nabla L(\mathbf{w}) \nabla L(\mathbf{w})^{\mathrm{T}}] = \hat{A} \Sigma \hat{A}$. Therefore we obtain

$$C = \mathbb{E}_{\mathbf{w}}[C(\mathbf{w})] = \frac{2L_{\mathrm{train}}}{S} \hat{A} - \frac{1}{S} \hat{A} \Sigma \hat{A}. \tag{73}$$

Now, we find $L_{\text{train}}$. First, we define $X \in \mathbb{R}^{N \times D}$ as $X_{ik} = (x_i)_k$, and $\vec{\epsilon} \in \mathbb{R}^N$ as $\vec{\epsilon}_i = \epsilon_i$. Then $\mathbf{w}^* = (1 - \Pi)\mathbf{u} + \hat{A}^+\mathbf{v} = (1 - \Pi)\mathbf{u} + (X^{\mathrm{T}}X)^+X^{\mathrm{T}}\vec{\epsilon}$.

With this notation, we have $\hat{A} = X^{\mathrm{T}}X/N$, and the loss function is expressed as

$$L(w) = \frac{1}{2}(\mathbf{w} - \mathbf{w}^*)^{\mathrm{T}}\hat{A}(\mathbf{w} - \mathbf{w}^*) - \frac{1}{2N}\vec{\epsilon}^{\mathrm{T}}X(X^{\mathrm{T}}X)^+X^{\mathrm{T}}\vec{\epsilon} + \frac{1}{2N}\sum_{i=1}^{N}\epsilon_i^2. \tag{74}$$

We therefore obtain

$$L_{\text{train}} = \frac{1}{2}\text{Tr}[\hat{A}\Sigma] - \frac{1}{2N}\mathbb{E}[\vec{\epsilon}^{\mathrm{T}}X(X^{\mathrm{T}}X)^+X^{\mathrm{T}}\vec{\epsilon}] + \frac{\sigma^2}{2}. \tag{75}$$

Here,

$$\mathbb{E}[\vec{\epsilon}^{\mathrm{T}}X(X^{\mathrm{T}}X)^+X^{\mathrm{T}}\vec{\epsilon}] = \sigma^2\text{Tr}[(X^{\mathrm{T}}X)(X^{\mathrm{T}}X)^+] = \sigma^2\text{Tr}(1 - \Pi). \tag{76}$$

We can prove that the following identity is almost surely satisfied (Hastie et al., 2019) as long as the smallest eigenvalue of $A$ (not $\hat{A}$) is positive:

$$\text{Tr}(1 - \Pi) = \min\{D, N\}. \tag{77}$$

We therefore obtain

$$L_{\text{train}} = \frac{1}{2}\text{Tr}[\hat{A}\Sigma] - \frac{\sigma^2}{2N}\min\{D, N\} + \frac{\sigma^2}{2} = \begin{cases} \frac{1}{2}\text{Tr}[\hat{A}\Sigma] + \frac{1}{2}\left(1 - \frac{1}{\alpha}\right)\sigma^2 & \text{for } \alpha > 1, \\ \frac{1}{2}\text{Tr}[\hat{A}\Sigma] & \text{for } \alpha \leq 1 \end{cases} \tag{78}$$

By substituting Eq. (78) into Eq. (73), we obtain the following SGD noise covariance:

$$C = \begin{cases} \frac{1}{S}\left(\text{Tr}[\hat{A}\Sigma] - \hat{A}\Sigma\right)\hat{A} + \frac{\sigma^2}{S}\left(1 - \frac{1}{\alpha}\right)\hat{A} & \text{for } \alpha > 1, \\ \frac{1}{S}\left(\text{Tr}[\hat{A}\Sigma] - \hat{A}\Sigma\right)\hat{A} & \text{for } \alpha \leq 1. \end{cases} \tag{79}$$

This finishes the proof. □

### D.1.2    Proof of Theorem 6

*Proof.* We have to solve this equation:

$$\hat{A}\Sigma + \Sigma\hat{A} - \lambda\hat{A}\Sigma\hat{A} = \lambda C, \tag{80}$$

where $C$ is given in Proposition 4. Using the similar trick of multiplying by $\hat{G} := 2I_D - \lambda\left(1 - \frac{1}{S}\right)\hat{A}$ as in Appendix E.2.2, one obtains

$$\text{Tr}[\hat{A}\Sigma] = \begin{cases} \frac{\lambda\sigma^2}{S}\left(1 - \frac{1}{\alpha}\right)\hat{\kappa} & \text{for } \alpha > 1; \\ 0 & \text{for } \alpha \leq 1, \end{cases} \tag{81}$$

where $\hat{\kappa} := \frac{\text{Tr}[\hat{G}^{-1}\hat{A}]}{1 - \frac{\lambda}{S}\text{Tr}[\hat{G}^{-1}\hat{A}]}$ with $\hat{G} := 2I_D - \lambda\left(1 - \frac{1}{S}\right)\hat{A}$.

Substituting the above trace into the matrix equation, we have

$$\Sigma = \begin{cases} \frac{\lambda\sigma^2}{S}\left(1 - \frac{1}{\alpha}\right)\left(1 + \frac{\lambda}{S}\hat{\kappa}\right)\hat{G}^{-1} & \text{for } \alpha > 1; \\ 0 & \text{for } \alpha \leq 1. \end{cases} \tag{82}$$

□

## D.2    Second-order Methods

**Proposition 5.** *Suppose that we run DNM with $\Lambda := \lambda A^{-1}$ with random noise in the label. The model fluctuation is*

$$\Sigma = \frac{\lambda\sigma^2}{gS - \lambda D}A^{-1}, \tag{83}$$

*where $g := 2(1 - \mu) - \left(\frac{1 - \mu}{1 + \mu} + \frac{1}{S}\right)\lambda$.*

*Proof.* Substituting $\Lambda = \lambda A^{-1}$ into Eqs. (3) and (5) yields

$$g\Sigma = \frac{\lambda}{S}\left(\mathrm{Tr}[A\Sigma] + \sigma^2\right)A^{-1}, \tag{84}$$

where $g :== 2(1 - \mu) - \left(\frac{1-\mu}{1+\mu} + \frac{1}{S}\right)\lambda$. Multiplying $A$ and taking trace on both sides, we have

$$\mathrm{Tr}[A\Sigma] = \frac{\lambda D\sigma^2}{gS - \lambda D}. \tag{85}$$

Therefore, the model fluctuation is

$$\Sigma = \frac{\lambda\sigma^2}{gS - \lambda D}A^{-1}. \tag{86}$$

$\square$

**Proposition 6.** *Suppose that we run NGD with $\Lambda := \frac{\lambda}{S}J(\mathbf{w})^{-1} \approx \frac{\lambda}{S}C^{-1}$ with random noise in the label. The model fluctuation is*

$$\Sigma = \left[\frac{\lambda}{4}g - \frac{1}{2}\frac{\sigma^2}{1+D} + \frac{1}{4}\sqrt{\lambda^2 g^2 + 4\lambda\left(g - \frac{2}{1+D}\frac{1}{1+\mu}\right)\frac{\sigma^2}{1+D} + 4\left(\frac{\sigma^2}{1+D}\right)^2}\right]A^{-1}, \tag{87}$$

*where $g := \frac{1}{1+D}\frac{1}{1+\mu} + \frac{1}{1-\mu}\frac{1}{S}$.*

*Proof.* Similarly to the previous case, the matrix equation satisfied by $\Sigma$ is

$$(1-\mu)(C^{-1}A\Sigma + \Sigma AC^{-1}) - \frac{1+\mu^2}{1-\mu^2}\frac{\lambda}{S}C^{-1}A\Sigma AC^{-1} + \frac{\mu}{1-\mu^2}\frac{\lambda}{S}(C^{-1}AC^{-1}A\Sigma + \Sigma AC^{-1}AC^{-1}) = \frac{\lambda}{S}C^{-1}. \tag{88}$$

Although it is not obvious how to directly solve this equation, it is possible to guess one solution according to the hope that $\Sigma$ be proportional to $J^{-1}$, in turn, $A^{-1}$ (Amari, 1998; Liu et al., 2021). We assume that $\Sigma = xA^{-1}$ and substitute it into the above equation to solve for $x$. This yields one solution without claiming its uniqueness. By simple algebra, this $x$ is solved to be

$$x = \frac{\lambda}{4}g - \frac{1}{2}\frac{\sigma^2}{1+D} + \frac{1}{4}\sqrt{\lambda^2 g^2 + 4\lambda\left(g - \frac{2}{1+D}\frac{1}{1+\mu}\right)\frac{\sigma^2}{1+D} + 4\left(\frac{\sigma^2}{1+D}\right)^2}. \tag{89}$$

Let $\sigma = 0$. We obtain the result in Sec. 6.4. $\square$

### D.3 ESTIMATION OF TAIL INDEX

In Mori et al. (2021); Meng et al. (2020), it is shown that the (1d) discrete-time SGD results in a distribution that is similar to a Student's t-distribution:

$$p(w) \sim (\sigma^2 + aw^2)^{-\frac{1+\beta}{2}}, \tag{90}$$

where $\sigma^2$ is the degree of noise in the label, and $a$ is the local curvature of the minimum. For large $w$, this distribution is a power-law distribution with tail index:

$$p(|w|) \sim |w|^{-(1+\beta)}, \tag{91}$$

and it is not hard to check that $\beta$ also equal to the smallest moment of $w$ that diverges: $\mathbb{E}[w^\beta] = \infty$. Therefore, estimating $\beta$ can be of great use both empirically and theoretically.

In continuous-time, it is found that $\beta_{\mathrm{cts}} = \frac{2S}{a\lambda} + 1$ (Mori et al., 2021). For discrete-time SGD, we hypothesize that the discrete-time nature causes a change in the tail index $\beta = \beta_{\mathrm{cts}} + \epsilon$, and we are interested in finding $\epsilon$. We propose a "semi-continuous" approximation to give the formula to estimate the tail index. Notice that Theorem 2 gives the variance of the discrete-time SGD, while Eq. (90) can be integrated to give another value of the variance, and the two expressions must be equal for consistency. This gives us an equation that $\beta$ must satisfy:

$$\int p(w;\beta)(w - \mathbb{E}[w])^2 = \mathrm{Var}[w], \tag{92}$$

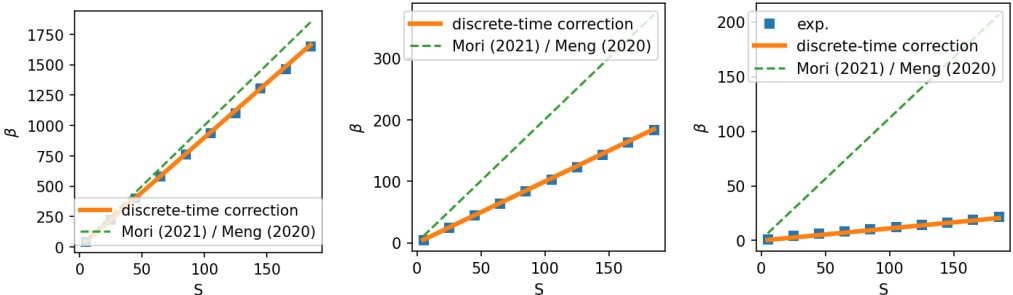

Figure 7: Tail index $\beta$ of the stationary distribution of SGD in a 1d linear regression problem. **Left to Right**: $a\lambda = 0.2,\ 1.0,\ 1.8$.

this procedure gives the following formula:

$$\beta(\lambda, S) = \frac{2S}{a\lambda} - S = \beta_{\text{cts}} + \epsilon, \tag{93}$$

and one immediately recognizes that $-(S+1)$ is the discrete-time contribution to the tail index. See Figure 7 for additional experiments. We see that the proposed formula agrees with the experimentally measured value of the tail index for all ranges of the learning rate, while the result of Mori et al. (2021) is only correct when $\lambda \to 0^+$. Hodgkinson and Mahoney (2020) also studies the tail exponent of discrete-time SGD; however, their conclusion is only that the "index decreases with the learning rate and increases with the batch size". In contrast, our result give the functional form of the tail index directly. In fact, this is the first work that gives any functional form for the tail index of discrete-time SGD fluctuation to the best of our knowledge.

The following proposition gives the intermediate steps in the calculation.

**Proposition 7.** (*Tail index estimation for discrete-time SGD*) *Let the parameter distribution be* $p(w) \sim \left(\sigma^2 + aw^2\right)^{-\frac{1+\beta}{2}}$, *and Var$[w]$ be given by Theorem 2. Then*

$$\beta(\lambda, S) = \frac{2S}{a\lambda} - S. \tag{94}$$

*Proof.* The normalization factor for the distribution exists if $\beta > 0$:

$$\mathcal{N} = \sqrt{\frac{a}{\pi}} \frac{\sigma^\beta \Gamma(\frac{1+\beta}{2})}{\Gamma(\frac{\beta}{2})}. \tag{95}$$

If $\beta > 2$, the variance exists and the value is

$$\text{Var}[w] = \frac{\sigma^2}{a(\beta - 2)}. \tag{96}$$

By equating Eq. (96) with the exact variance (6), we are able to solve for an expression of the tail index as

$$\beta(\lambda, S) = \frac{2S}{a\lambda} - S. \tag{97}$$

$\square$

# E  PROOFS AND ADDITIONAL THEORETICAL CONSIDERATIONS

## E.1  PROOF OF PROPOSITION 1

The with-replacement sampling is defined in Definition 2. Let us here define the without-replacement sampling.

**Definition 3.** A minibatch SGD *without replacement* computes the update to the parameter $\mathbf{w}$ with the following set of equations:

$$\begin{cases} \hat{\mathbf{g}}_t = \frac{1}{S} \sum_{i \in B_t} \nabla \ell(x_i, y_i, \mathbf{w}_{t-1}); \\ \mathbf{w}_t = \mathbf{w}_{t-1} - \lambda \hat{\mathbf{g}}_t, \end{cases} \tag{98}$$

where $S := |B_t| \leq N$ is the minibatch size, and the set $B_t$ is an element uniformly-randomly drawn from the set of all $S$-size subsets of $\{1, ..., N\}$.

From the definition of the update rule for sampling with or without replacement, the covariance matrix of the SGD noise can be exactly derived.

**Proposition 8.** *The covariance matrices of noise in SGD due to minibatch sampling as defined in Definitions 2 and 3 with an arbitrary $N$ are*

$$C(\mathbf{w}) = \begin{cases} \frac{1}{S} \left[ \frac{1}{N} \sum_{i=1}^{N} \nabla \ell_i \nabla \ell_i^{\mathrm{T}} - \nabla L(\mathbf{w}) \nabla L(\mathbf{w})^{\mathrm{T}} \right], & \text{(with replacement)} \\ \frac{N-S}{S(N-1)} \left[ \frac{1}{N} \sum_{i=1}^{N} \nabla \ell_i \nabla \ell_i^{\mathrm{T}} - \nabla L(\mathbf{w}) \nabla L(\mathbf{w})^{\mathrm{T}} \right], & \text{(without replacement)} \end{cases} \tag{99}$$

*where the shorthand notation $\ell_i(\mathbf{w}) := l(x_i, y_i, \mathbf{w})$ is used.*

In the limit of $S = 1$ or $N \gg S$, two cases coincide. In the $N \gg S$ limit, both methods of sampling have the same noise covariance as stated in Proposition 1:

$$C(\mathbf{w}) = \frac{1}{SN} \sum_{i=1}^{N} \nabla \ell_i \nabla \ell_i^{\mathrm{T}} - \frac{1}{S} \nabla L(\mathbf{w}) \nabla L(\mathbf{w})^{\mathrm{T}}. \tag{100}$$

**Remark.** *We also note that a different way of defining minibatch noise exists in Hoffer et al. (2017). The difference is that our definition requires the size of each minibatch to be exactly $S$, while Hoffer et al. (2017) treats the batch size also as a random variable and is only expected to be $S$. In comparison, our definition agrees better with the common practice.*

Now we prove Proposition 8.

*Proof.* We derive the noise covariance matrices for sampling with and without replacement. We first derive the case with replacement. According to the definition, the stochastic gradient for sampling with replacement can be rewritten as

$$\hat{\mathbf{g}} = \frac{1}{S} \sum_{n=1}^{N} \mathbf{g}_n s_n, \tag{101}$$

where $\mathbf{g}_n := \nabla \ell_n$ and

$$s_n = l, \text{ if } l - \text{multiple } n's \text{ are sampled in S, with } 0 \leq l \leq S. \tag{102}$$

The probability of $s_n$ assuming value $l$ is given by the multinomial distribution

$$P(s_n = l) = \binom{S}{l} \left(\frac{1}{N}\right)^l \left(1 - \frac{1}{N}\right)^{S-l}. \tag{103}$$

Therefore, the expectation value of $s_n$ is given by

$$\mathbb{E}_{\mathrm{B}}[s_n] = \sum_{l=0}^{S} l P(s_n = l) = \frac{S}{N}, \tag{104}$$

which gives

$$\mathbb{E}_{\mathrm{B}}[\hat{\mathbf{g}}] = \mathbf{g} := \frac{1}{N} \sum_{n=1}^{N} \mathbf{g}_n = \nabla L(\mathbf{w}). \tag{105}$$

For the covariance, we first calculate the covariance between $s_n$ and $s_{n'}$. Due to the properties of the covariance of multinomial distribution, we have for $n \neq n'$

$$\mathbb{E}_{\mathrm{B}}[s_n s_{n'}] = \mathrm{cov}[s_n, s_{n'}] + \mathbb{E}[s_n]^2$$
$$= -\frac{S}{N^2} + \frac{S^2}{N^2}$$
$$= \frac{S(S-1)}{N^2}; \tag{106}$$

and for $n = n'$

$$\mathbb{E}_{\mathrm{B}}[s_n s_n] = \mathrm{Var}[s_n] + \mathbb{E}[s_n]^2$$
$$= \frac{S}{N}\frac{N-1}{N} + \frac{S^2}{N^2}$$
$$= \frac{SN + S(S-1)}{N^2}. \tag{107}$$

Substituting these results into the definition of the noise covariance yields

$$C(\mathbf{w}) = \mathbb{E}_{\mathrm{B}}[\hat{\mathbf{g}}\hat{\mathbf{g}}^{\mathrm{T}}] - \mathbb{E}_{\mathrm{B}}[\hat{\mathbf{g}}]\mathbb{E}_{\mathrm{B}}[\hat{\mathbf{g}}]^{\mathrm{T}}$$
$$= \frac{1}{S^2}\sum_{n=1}^{N}\sum_{n'=1}^{N}\mathbf{g}_n\mathbf{g}_{n'}^{\mathrm{T}}\mathbb{E}_{\mathrm{B}}[s_n s_{n'}] - \mathbf{g}\mathbf{g}^{\mathrm{T}}$$
$$= \frac{1}{S^2}\sum_{n,n'=1}^{N}\mathbf{g}_n\mathbf{g}_{n'}^{\mathrm{T}}\frac{S(S-1)}{N^2} + \frac{1}{S^2}\sum_{n=1}^{N}\mathbf{g}_n\mathbf{g}_n^{\mathrm{T}}\left[\frac{SN+S(S-1)}{N^2} - \frac{S(S-1)}{N^2}\right] - \mathbf{g}\mathbf{g}^{\mathrm{T}}$$
$$= \frac{1}{NS}\sum_{n=1}^{N}\mathbf{g}_n\mathbf{g}_n^{\mathrm{T}} - \frac{1}{S}\mathbf{g}\mathbf{g}^{\mathrm{T}}$$
$$= \frac{1}{S}\left[\frac{1}{N}\sum_{i=1}^{N}\nabla\ell_i\nabla\ell_i^{\mathrm{T}} - \nabla L(\mathbf{w})\nabla L(\mathbf{w})^{\mathrm{T}}\right]. \tag{108}$$

Then, we derive the noise covariance for sampling without replacement. Similarly, according to the definition, the stochastic gradient for sampling without replacement can be rewritten as

$$\hat{\mathbf{g}} = \frac{1}{S}\sum_{n=1}^{N}\mathbf{g}_n s_n, \tag{109}$$

where

$$s_n = \begin{cases} 0, \text{if } n \notin S; \\ 1, \text{if } n \in S. \end{cases} \tag{110}$$

The probability of $n$ that is sampled in $S$ from $N$ is given by

$$P(s_n = 1) = \frac{\binom{N-1}{S-1}}{\binom{N}{S}} = \frac{S}{N}. \tag{111}$$

The expectation value of $s_n$ is then given by

$$\mathbb{E}_{\mathrm{B}}[s_n] = P(s_n = 1) = \frac{S}{N}, \tag{112}$$

which gives

$$\mathbb{E}_{\mathrm{B}}[\hat{\mathbf{g}}] = \mathbf{g} := \frac{1}{N}\sum_{n=1}^{N}\mathbf{g}_n = \nabla L(\mathbf{w}). \tag{113}$$

For the covariance, we first calculate the covariance between $s_n$ and $s_{n'}$. By definition, we have for $n \neq n'$

$$\mathbb{E}_{\mathrm{B}}[s_n s_{n'}] = P(s_n = 1, s_n' = 1) = P(s_n = 1 | s_n' = 1)P(s_n' = 1)$$
$$= \frac{\binom{N-2}{S-2}}{\binom{N-1}{S-1}}\frac{\binom{N-1}{S-1}}{\binom{N}{S}} = \frac{S(S-1)}{N(N-1)}; \tag{114}$$

and for $n = n'$

$$\mathbb{E}_{\mathrm{B}}[s_n s_n] = P(s_n = l) = \frac{S}{N}. \tag{115}$$

Substituting these results into the definition of the noise covariance yields

$$
\begin{aligned}
C(\mathbf{w}) &= \mathbb{E}_{\mathrm{B}}[\hat{\mathbf{g}}\hat{\mathbf{g}}^{\mathrm{T}}] - \mathbb{E}_{\mathrm{B}}[\hat{\mathbf{g}}]\mathbb{E}_{\mathrm{B}}[\hat{\mathbf{g}}]^{\mathrm{T}} \\
&= \frac{1}{S^2} \sum_{n=1}^{N} \sum_{n'=1}^{N} \mathbf{g}_n \mathbf{g}_{n'}^{\mathrm{T}} \mathbb{E}_{\mathrm{B}}[s_n s_{n'}] - \mathbf{g}\mathbf{g}^{\mathrm{T}} \\
&= \frac{1}{S^2} \sum_{n,n'=1}^{N} \mathbf{g}_n \mathbf{g}_{n'}^{\mathrm{T}} \frac{S(S-1)}{N(N-1)} + \frac{1}{S^2} \sum_{n=1}^{N} \mathbf{g}_n \mathbf{g}_n^{\mathrm{T}} \left[ \frac{S}{N} - \frac{S(S-1)}{N(N-1)} \right] - \mathbf{g}\mathbf{g}^{\mathrm{T}} \\
&= \frac{1}{NS} \frac{N-S}{N-1} \sum_{n=1}^{N} \mathbf{g}_n \mathbf{g}_n^{\mathrm{T}} - \frac{N-S}{S(N-1)} \mathbf{g}\mathbf{g}^{\mathrm{T}} \\
&= \frac{N-S}{S(N-1)} \left[ \frac{1}{N} \sum_{i=1}^{N} \nabla \ell_i \nabla \ell_i^{\mathrm{T}} - \nabla L(\mathbf{w}) \nabla L(\mathbf{w})^{\mathrm{T}} \right]. \tag{116}
\end{aligned}
$$

$\square$

## E.2 Proofs in Sec. 4.2

### E.2.1 Proof of Lemma 1

*Proof.* From the definition of noise covariance (2), the covariance matrix for the noise in the label is

$$
\begin{aligned}
C(\mathbf{w}) &= \frac{1}{NS} \sum_{i=1}^{N} \nabla l_i(\mathbf{w}_{t-1}) \nabla l_i(\mathbf{w}_{t-1})^{\mathrm{T}} - \frac{1}{S} \nabla L(\mathbf{w}_{t-1}) \nabla L(\mathbf{w}_{t-1})^{\mathrm{T}} \\
&= \frac{1}{S} \frac{1}{N} \sum_i^N (\mathbf{w}^{\mathrm{T}} x_i - \epsilon_i) x_i x_i^{\mathrm{T}} (\mathbf{w}^{\mathrm{T}} x_i - \epsilon_i)^{\mathrm{T}} - \frac{1}{S} \left[ \frac{1}{N} \sum_i^N (\mathbf{w}^{\mathrm{T}} x_i - \epsilon_i) x_i \right] \left[ \frac{1}{N} \sum_j^N x_j^{\mathrm{T}} (\mathbf{w}^{\mathrm{T}} x_j - \epsilon_j)^{\mathrm{T}} \right] \\
&= \frac{1}{S} \frac{1}{N} \sum_i^N (\mathbf{w}^{\mathrm{T}} x_i x_i x_i^{\mathrm{T}} x_i^{\mathrm{T}} \mathbf{w} + \epsilon_i^2 x_i x_i^{\mathrm{T}}) - \frac{1}{S} \left[ \frac{1}{N} \sum_i^N (\mathbf{w}^{\mathrm{T}} x_i x_i) \right] \left[ \frac{1}{N} \sum_j^N (x_i^{\mathrm{T}} x_i^{\mathrm{T}} \mathbf{w}) \right] \tag{117} \\
&= \frac{1}{S} (A \mathbf{w}\mathbf{w}^{\mathrm{T}} A + \mathrm{Tr}[A \mathbf{w}\mathbf{w}^{\mathrm{T}}] A + \sigma^2 A), \tag{118}
\end{aligned}
$$

where we have invoked the law of large numbers and the expectation value of the product of four Gaussian random variables in the third line is evaluated as follows.

Because $N$ is large, we invoke the law of large numbers to obtain the $(j, k)$-th component of the matrix as

$$\lim_{N \to \infty} \frac{1}{N} \sum_{i=1}^{N} (\mathbf{w}^{\mathrm{T}} x_i x_i x_i^{\mathrm{T}} x_i^{\mathrm{T}} \mathbf{w})_{jk} = \mathbb{E}_{\mathrm{B}}[\mathbf{w}^{\mathrm{T}} x x x^{\mathrm{T}} x^{\mathrm{T}} \mathbf{w}]_{jk} = \mathbb{E}_{\mathrm{B}} \left[ \sum_i^D w_i x_i x_j x_k \sum_{i'}^D x_{i'} w_{i'} \right]. \tag{119}$$

Because the average is taken with respect to $x$ and each $x$ is a Gaussian random variable, we apply the expression for the product of four Gaussian random variables $\mathbb{E}[x_1 x_2 x_3 x_4] = \mathbb{E}[x_1 x_2]\mathbb{E}[x_3 x_4] + \mathbb{E}[x_1 x_3]\mathbb{E}[x_2 x_4] + \mathbb{E}[x_1 x_4]\mathbb{E}[x_2 x_3] - 2\mathbb{E}[x_1]\mathbb{E}[x2]\mathbb{E}[x_3]\mathbb{E}[x_4]$ (Janssen and Stoica, 1988) to obtain

$$
\begin{aligned}
&\mathbb{E}_{\mathrm{B}} \left[ \sum_i^D w_i x_i x_j x_k \sum_{i'}^D x_{i'} w_{i'} \right] \\
&= \mathbb{E}_{\mathrm{B}} \left[ \sum_i^D w_i x_i x_j \right] \mathbb{E}_{\mathrm{B}} \left[ x_k \sum_{i'}^D x_{i'} w_{i'} \right] + \mathbb{E}_{\mathrm{B}} \left[ \sum_i^D w_i x_i x_k \right] \mathbb{E}_{\mathrm{B}} \left[ x_j \sum_{i'}^D x_{i'} w_{i'} \right] \\
&\quad + \mathbb{E}_{\mathrm{B}} \left[ \sum_i^D w_i x_i \sum_{i'}^D x_{i'} w_{i'} \right] \mathbb{E}_{\mathrm{B}} [x_j x_k] \\
&= 2(A \mathbf{w}\mathbf{w}^{\mathrm{T}} A)_{jk} + \mathrm{Tr}[A \mathbf{w}\mathbf{w}^{\mathrm{T}}] A_{jk}. \tag{120}
\end{aligned}
$$

Writing $\Sigma := \mathbb{E}_{\mathbf{w}}[\mathbf{w}\mathbf{w}^{\mathrm{T}}]$, we obtain

$$\mathbb{E}_{\mathbf{w}}[C(\mathbf{w})] =: C = \frac{1}{S} (A\Sigma A + \mathrm{Tr}[A\Sigma] A + \sigma^2 A). \tag{121}$$

$\square$

This method has been utilized repeatedly in this work.

### E.2.2 Proof of Theorem 2

*Proof.* We substitute Eq. (5) into Eq. (3) which is a general solution obtained in a recent work (Liu et al., 2021):

$$(1 - \mu)(\Lambda A\Sigma + \Sigma A\Lambda) - \frac{1 + \mu^2}{1 - \mu^2}\Lambda A\Sigma A\Lambda + \frac{\mu}{1 - \mu^2}(\Lambda A\Lambda A\Sigma + \Sigma A\Lambda A\Lambda) = \Lambda C\Lambda. \tag{122}$$

To solve it, we assume the commutation relation that $[\Lambda, A] := \Lambda A - A\Lambda = 0$. Therefore, the above equation can be alternatively rewritten as

$$\left[(1 - \mu)I_D - \frac{1}{2}\left(\frac{1 - \mu}{1 + \mu} + \frac{1}{S}\right)\Lambda A\right]\Sigma A\Lambda + \Lambda A\Sigma\left[(1 - \mu)I_D - \frac{1}{2}\left(\frac{1 - \mu}{1 + \mu} + \frac{1}{S}\right)\Lambda A\right]$$
$$- \frac{1}{S}\mathrm{Tr}[A\Sigma]\Lambda A = \frac{1}{S}\sigma^2\Lambda A. \tag{123}$$

To solve this equation, we first need to solve for $\mathrm{Tr}[A\Sigma]$. Multiplying Eq. (123) by $G_\mu^{-1} :=$ $\left[2(1 - \mu)I_D - \left(\frac{1 - \mu}{1 + \mu} + \frac{1}{S}\right)\Lambda A\right]^{-1}$ and taking trace, we obtain

$$\mathrm{Tr}[A\Sigma] - \frac{1}{S}\mathrm{Tr}[A\Sigma]\mathrm{Tr}[\Lambda AG_\mu^{-1}] = \frac{1}{S}\sigma^2\mathrm{Tr}[\Lambda AG_\mu^{-1}], \tag{124}$$

which solves to give

$$\mathrm{Tr}[A\Sigma] = \frac{\sigma^2}{S}\frac{\mathrm{Tr}[\Lambda AG_\mu^{-1}]}{1 - \frac{1}{S}\mathrm{Tr}[\Lambda AG_\mu^{-1}]} := \frac{\sigma^2}{S}\kappa_\mu. \tag{125}$$

Therefore, $\Sigma$ is

$$\Sigma = \frac{\sigma^2}{S}\left(1 + \frac{\kappa}{S}\right)\Lambda\left[2(1 - \mu)I_D - \left(\frac{1 - \mu}{1 + \mu} + \frac{1}{S}\right)\Lambda A\right]^{-1}. \tag{126}$$

$\square$

### E.2.3 Training Error and Test Error for Label Noise

In the following theorem, we calculate the expected training and test loss for random noise in the label.

**Theorem 8.** (*Approximation error and test loss for SGD noise in the label*) *The expected approximation error, or the training loss, is defined as $L_{\mathrm{train}} := \mathbb{E}_{\mathbf{w}}[L(\mathbf{w})]$; the expected test loss is defined as $L_{\mathrm{test}} := \frac{1}{2}\mathbb{E}_{\mathbf{w}}\mathbb{E}_{\mathrm{B}}\left[(\mathbf{w}^{\mathrm{T}}x)^2\right]$. For SGD with noise in the label given by Eq. (5), the expected approximation error and test loss are*

$$L_{\mathrm{train}} = \frac{\sigma^2}{2}\left(1 + \frac{\lambda\kappa}{S}\right), \tag{127}$$

$$L_{\mathrm{test}} = \frac{\lambda\sigma^2}{2S}\kappa. \tag{128}$$

**Remark.** *Notably, the training loss decomposes into two additive terms. The term that is proportional to $1$ is the bias, caused by insufficient model expressivity to perfectly fit all the data points, while the second term that is proportional to $\lambda\kappa/S$ is the variance in the model parameter, induced by the randomness of minibatch noise.*

**Remark.** *When the learning rate $\lambda$ is vanishingly small, the expected test loss diminishes whereas the training error remains finite as long as label noise exists.*

*Proof.* We first calculate the approximation error. By definition,

$$L_{\mathrm{train}} := \mathbb{E}_{\mathbf{w}}[L(\mathbf{w})]$$
$$= \frac{1}{2}\mathrm{Tr}[A\Sigma] + \frac{1}{2}\sigma^2 = \frac{1}{2}\frac{\lambda\sigma^2}{S}\kappa + \frac{1}{2}\sigma^2$$
$$= \frac{\sigma^2}{2}\left(1 + \frac{\lambda\kappa}{S}\right). \tag{129}$$

The test loss is

$$L_{\text{test}} = \frac{1}{2}\mathbb{E}_{\mathbf{w}}\left[\mathbf{w}^{\text{T}}A\mathbf{w}\right] = \frac{1}{2}\text{Tr}[A\Sigma]$$
$$= \frac{\lambda\sigma^2}{2S}\kappa. \tag{130}$$

$\square$

### E.3 MINIBATCH NOISE FOR RANDOM NOISE IN THE INPUT

#### E.3.1 NOISE STRUCTURE

Similar to label noise, noise in the input data can also cause fluctuation. We assume that the training data points $\tilde{x}_i = x_i + \eta_i$ can be decomposed into a signal part and a random part. As before, we assume Gaussian distributions, $x_i \sim \mathcal{N}(0, A)$ and $\eta_i \sim \mathcal{N}(0, B)$. The problem remains analytically solvable if we replace the Gaussian assumption by the weaker assumption that the fourth-order moment exists and takes some matrix form. For conciseness, we assume that there is no noise in the label, namely $y_i = \mathbf{u}^{\text{T}}x_i$ with a constant vector $\mathbf{u}$. One important quantity in this case will be $\mathbf{u}\mathbf{u}^{\text{T}} := U$. Notice that the trick $\mathbf{w} - \mathbf{u} = \mathbf{w}$ no more works, and so we write the difference explicitly here. The loss function then takes the form

$$L(\mathbf{w}) = \frac{1}{2N}\sum_{i=1}^{N}\left[(\mathbf{w} - \mathbf{u})^{\text{T}}x_i + \mathbf{w}^{\text{T}}\eta_i\right]^2 = \frac{1}{2}(\mathbf{w} - \mathbf{u})^{\text{T}}A(\mathbf{w} - \mathbf{u}) + \frac{1}{2}\mathbf{w}^{\text{T}}B\mathbf{w}. \tag{131}$$

The gradient $\nabla L = (A + B)(\mathbf{w} - \mathbf{u}) + B\mathbf{u}$ vanishes at $\mathbf{w}_* := (A + B)^{-1}A\mathbf{u}$, which is the minimum of the loss function and the expectation of the parameter at convergence. It can be seen that, even at the minimum $\mathbf{w}_*$, the loss function remains finite unless $\mathbf{u} = 0$, which reflects the fact that in the presence of input noise, the network is not expressive enough to memorize all the information of the data. The SGD noise covariance for this type of noise is calculated in the following proposition.

**Proposition 9.** (*Covariance matrix for SGD noise in the input*) *Let the algorithm be updated according to Eq. (1) or (98) with random noise in the input while the limit $N \to \infty$ is taken with $D$ held fixed. Then the noise covariance is*

$$C = \frac{1}{S}\left\{K\Sigma K + \text{Tr}[K\Sigma]K + \text{Tr}[AK^{-1}BU]K\right\}, \tag{132}$$

*where $K := A + B$, and $\Sigma := \mathbb{E}_{\mathbf{w}}\left[(\mathbf{w} - \mathbf{w}_*)(\mathbf{w} - \mathbf{w}_*)^{\text{T}}\right]$.*

**Remark.** *It can be seen that the form of the covariance (132) of input noise is similar to that of label noise (5) with replacing $A$ by $K$ and $\sigma^2$ by $\text{Tr}[AK^{-1}BU]$, suggesting that these two types of noise share a similar nature.*

Defining the test loss as $L_{\text{test}} := \frac{1}{2}\mathbb{E}_{\mathbf{w}}\mathbb{E}_{\text{B}}\left[(\mathbf{w}^{\text{T}}x - \mathbf{u}^{\text{T}}x)^2\right]$, Proposition 9 can then be used to calculate the test loss and the model fluctuation.

**Theorem 9.** (*Training error, test loss and model fluctuation for noise in the input*) *The expected training loss is defined as $L_{\text{train}} := \mathbb{E}_{\mathbf{w}}[L(\mathbf{w})]$, and the expected test loss is defined as $L_{\text{test}} := \frac{1}{2}\mathbb{E}_{\mathbf{w}}\mathbb{E}_{\text{B}}\left[(\mathbf{w}^{\text{T}}x - \mathbf{u}^{\text{T}}x)^2\right]$. For SGD with noise in the input given in Proposition (9), the expected approximation error and test loss are*

$$L_{\text{train}} = \frac{1}{2}\text{Tr}[AK^{-1}BU]\left(1 + \frac{\lambda}{S}\kappa'\right), \tag{133}$$

$$L_{\text{test}} = \frac{\lambda}{2S}\text{Tr}[AK^{-1}BU]\kappa' + \frac{1}{2}\text{Tr}[B'^{\text{T}}AB'U], \tag{134}$$

*where $\kappa' := \frac{\text{Tr}[KG'^{-1}]}{1 - \lambda\frac{1}{S}\text{Tr}[KG'^{-1}]}$ with $G' := 2I_D - \lambda\left(1 + \frac{1}{S}\right)K$, and $B' := K^{-1}B$. Moreover, let $[K, U] = 0$. Then the covariance matrix of model parameters is*

$$\Sigma = \frac{\lambda\text{Tr}[AK^{-1}BU]}{S}\left(1 + \frac{\lambda\kappa'}{S}\right)\left[2I_D - \lambda\left(1 + \frac{1}{S}\right)K\right]^{-1}. \tag{135}$$

**Remark.** *Note that $[K, U] = 0$ is necessary only for an analytical expression of $\Sigma$. It can be obtained by solving Eq. (144) even without invoking $[K, U] = 0$. In general, the condition that $[K, U] = 0$ does not hold. Therefore, only the training and test error can be calculated exactly.*

**Remark.** *The test loss is always smaller than or equal to the training loss because all matrices involved here are positive semidefinite.*

### E.3.2  PROOF OF PROPOSITION 9

*Proof.* We define $\Sigma := \mathbb{E}_{\mathbf{w}}\left[(\mathbf{w} - \mathbf{w}_*)(\mathbf{w} - \mathbf{w}_*)^{\mathrm{T}}\right]$. Then,

$$\mathbb{E}_{\mathbf{w}}[\mathbf{w}\mathbf{w}^{\mathrm{T}}] = \Sigma + (A + B)^{-1}AUA(A + B)^{-1} := \Sigma + A'UA'^{\mathrm{T}} := \Sigma_A, \tag{136}$$

$$\mathbb{E}_{\mathbf{w}}\left[(\mathbf{w} - \mathbf{u})(\mathbf{w} - \mathbf{u})^{\mathrm{T}}\right] = \Sigma + B'UB'^{\mathrm{T}} := \Sigma_B, \tag{137}$$

where we use the shorthand notations $A' := (A + B)^{-1}A$, $B' := (A + B)^{-1}B$ and $\Sigma_A := \Sigma + A'UA'^{\mathrm{T}}$, $\Sigma_B := \Sigma + B'UB'^{\mathrm{T}}$. We remark that the covariance matrix $\Sigma$ here still satisfies the matrix equation (3) with the Hessian being $K := A + B$.

The noise covariance is

$$
\begin{aligned}
C(\mathbf{w}) &= \frac{1}{S}\frac{1}{N}\sum_{i}^{N}(\mathbf{w}^{\mathrm{T}}\tilde{x}_i - \mathbf{u}^{\mathrm{T}}x_i)\tilde{x}_i\tilde{x}_i^{\mathrm{T}}(\mathbf{w}^{\mathrm{T}}\tilde{x}_i - \mathbf{u}^{\mathrm{T}}x_i)^{\mathrm{T}} - \frac{1}{S}\nabla L(\mathbf{w})\nabla L(\mathbf{w})^{\mathrm{T}} \\
&= \frac{1}{S}\Big\{A(\mathbf{w} - \mathbf{u})(\mathbf{w} - \mathbf{u})^{\mathrm{T}}A + B\mathbf{w}\mathbf{w}^{\mathrm{T}}B + A(\mathbf{w} - \mathbf{u})\mathbf{w}^{\mathrm{T}}B + B\mathbf{w}(\mathbf{w} - \mathbf{u})^{\mathrm{T}}A \\
&\qquad + \mathrm{Tr}[A(\mathbf{w} - \mathbf{u})(\mathbf{w} - \mathbf{u})^{\mathrm{T}}]K + \mathrm{Tr}[B\mathbf{w}\mathbf{w}^{\mathrm{T}}]K\Big\}.
\end{aligned}
\tag{138}
$$

In Eq. (138), there are four terms without trace and two terms with trace. We first calculate the traceless terms. For the latter two terms, we have

$$\mathbb{E}_{\mathbf{w}}[(\mathbf{w} - \mathbf{u})\mathbf{w}^{\mathrm{T}}] = \Sigma - A'UB'^{\mathrm{T}}, \tag{139}$$

$$\mathbb{E}_{\mathbf{w}}[\mathbf{w}(\mathbf{w} - \mathbf{u})^{\mathrm{T}}] = \Sigma - B'UA'^{\mathrm{T}}. \tag{140}$$

Because $A' + B' = I_D$, after simple algebra the four traceless terms result in $2(A + B)\Sigma(A + B)$.

The two traceful terms add to $\mathrm{Tr}[A\Sigma_B + B\Sigma_A]K$. With the relation $AB' = BA'$, what inside the trace is

$$A\Sigma_B + B\Sigma_A = K\Sigma + AK^{-1}BU. \tag{141}$$

Therefore, the asymptotic noise is

$$
\begin{aligned}
C &:= \mathbb{E}_{\mathbf{w}}[C(\mathbf{w})] \\
&= \frac{1}{S}\left\{K\Sigma K + \mathrm{Tr}[A\Sigma_B + B\Sigma_A]K\right\} \tag{142} \\
&= \frac{1}{S}\left\{K\Sigma K + \mathrm{Tr}[K\Sigma]K + \mathrm{Tr}[AK^{-1}BU]K\right\}. \tag{143}
\end{aligned}
$$

$\square$

### E.3.3  PROOF OF THEOREM 9

*Proof.* The matrix equation satisfied by $\Sigma$ is

$$\Sigma K + K\Sigma - \lambda\left(1 + \frac{1}{S}\right)K\Sigma K = \frac{\lambda}{S}\left(\mathrm{Tr}[K\Sigma]K + \mathrm{Tr}[AK^{-1}BU]K\right). \tag{144}$$

By using a similar technique as in Appendix E.2.2, the trace $\mathrm{Tr}[K\Sigma]$ can be calculated to give

$$\mathrm{Tr}[K\Sigma] = \frac{\lambda\mathrm{Tr}[AK^{-1}BU]}{S}\kappa', \tag{145}$$

where $\kappa' := \frac{\mathrm{Tr}[KG'^{-1}]}{1 - \lambda\frac{1}{S}\mathrm{Tr}[KG'^{-1}]}$ with $G' := 2I_D - \lambda\left(1 + \frac{1}{S}\right)K$.

With Eq. (145), the training error and the test error can be calculated. The approximation error is

$$L_{\mathrm{train}} = \mathbb{E}_{\mathbf{w}}[L(\mathbf{w})] = \frac{1}{2}\mathrm{Tr}[A\Sigma_B + B\Sigma_A] = \frac{1}{2}\mathrm{Tr}[AK^{-1}BU]\left(1 + \frac{\lambda}{S}\kappa'\right). \tag{146}$$

The test loss takes the form of a bias-variance tradeoff:

$$
\begin{aligned}
L_{\text{test}} &= \frac{1}{2}\mathbb{E}_{\mathbf{w}}\mathbb{E}_{\text{B}}\left[(\mathbf{w}^{\text{T}}x - \mathbf{u}^{\text{T}}x)^2\right] = \frac{1}{2}\mathbb{E}_{\mathbf{w}}\left[(\mathbf{w}-\mathbf{u})^{\text{T}}A(\mathbf{w}-\mathbf{u})\right] = \frac{1}{2}\text{Tr}[A\Sigma_B] \\
&= \frac{\lambda}{2S}\text{Tr}[AK^{-1}BU]\left(1 + \frac{\lambda\kappa'}{S}\right)\text{Tr}[AG'^{-1}] + \frac{1}{2}\text{Tr}[B'^{\text{T}}AB'U] \\
&= \frac{\lambda}{2S}\text{Tr}[AK^{-1}BU]\kappa' + \frac{1}{2}\text{Tr}[B'^{\text{T}}AB'U].
\end{aligned}
\tag{147}
$$

Let $[K,U] = 0$. Then $\Sigma$ can be explicitly solved because it is a function of $K$ and $U$. Specifically,

$$
\Sigma = \frac{\lambda\text{Tr}[AK^{-1}BU]}{S}\left(1 + \frac{\lambda\kappa'}{S}\right)\left[2I_D - \lambda\left(1 + \frac{1}{S}\right)K\right]^{-1}.
\tag{148}
$$

$\square$

### E.4 PROOFS IN SEC. 4.3

#### E.4.1 PROOF OF PROPOSITION 3

*Proof.* The covariance matrix of the noise is

$$
\begin{aligned}
C(\mathbf{w}) &= \frac{1}{S}\frac{1}{N}\sum_i^N\left[(\mathbf{w}-\mathbf{u})^{\text{T}}x_ix_i + \Gamma\mathbf{w}\right]\left[x_i^{\text{T}}x_i^{\text{T}}(\mathbf{w}-\mathbf{u}) + \mathbf{w}^{\text{T}}\Gamma\right] - \frac{1}{S}\nabla L_\Gamma(\mathbf{w})\nabla L_\Gamma(\mathbf{w})^{\text{T}} \\
&= \frac{1}{S}\left\{A(\mathbf{w}-\mathbf{u})(\mathbf{w}-\mathbf{u})^{\text{T}}A + \text{Tr}[A(\mathbf{w}-\mathbf{u})(\mathbf{w}-\mathbf{u})^{\text{T}}]A\right\}.
\end{aligned}
\tag{149}
$$

Using a similar trick as in Appendix E.3.2, the asymptotic noise is

$$
C = \frac{1}{S}\left(A\Sigma A + \text{Tr}[A\Sigma]A + \text{Tr}[\Gamma'^{\text{T}}A\Gamma'U]A + \Gamma A'UA'\Gamma\right).
\tag{150}
$$

$\square$

#### E.4.2 PROOF OF THEOREM 4

Besides the test loss and the model fluctuation, we derive the approximation error here as well.

**Theorem.** (Training error, test loss and model fluctuation for learning with $L_2$ regularization) The expected training loss is defined as $L_{\text{train}} := \mathbb{E}_{\mathbf{w}}[L(\mathbf{w})]$, and the expected test loss is defined as $L_{\text{test}} := \frac{1}{2}\mathbb{E}_{\mathbf{w}}\mathbb{E}_{\text{B}}\left[(\mathbf{w}^{\text{T}}x - \mathbf{u}^{\text{T}}x)^2\right]$. For noise induced by $L_2$ regularization given in Proposition 3, let $[A,\Gamma] = 0$. Then the expected approximation error and test loss are

$$
\begin{aligned}
L_{\text{train}} &= \frac{\lambda}{2S}\text{Tr}[AK^{-2}\Gamma^2 U]\text{Tr}[AG^{-1}]\left(1 + \frac{\lambda\kappa}{S}\right) + \frac{\lambda}{2S}\left(\text{Tr}[A^2K^{-2}\Gamma^2 G^{-1}U] + \frac{\lambda r}{S}\text{Tr}[AG^{-1}]\right) \\
&\quad + \frac{1}{2}\text{Tr}[AK^{-1}\Gamma U],
\end{aligned}
\tag{151}
$$

$$
L_{\text{test}} = \frac{\lambda}{2S}\left(\text{Tr}[AK^{-2}\Gamma^2 U]\kappa + r\right) + \frac{1}{2}\text{Tr}[AK^{-2}\Gamma^2 U],
\tag{152}
$$

where $\kappa := \frac{\text{Tr}[A^2K^{-1}G^{-1}]}{1 - \frac{\lambda}{S}\text{Tr}[A^2K^{-1}G^{-1}]}$, $r := \frac{\text{Tr}[A^3K^{-3}\Gamma^2 G^{-1}U]}{1 - \frac{\lambda}{S}\text{Tr}[A^2K^{-1}G^{-1}]}$, with $G := 2I_D - \lambda\left(K + \frac{1}{S}K^{-1}A^2\right)$. Moreover, if $A$, $\Gamma$ and $U$ commute with each other, the model fluctuation is

$$
\Sigma = \frac{\lambda}{S}\text{Tr}[AK^{-2}\Gamma^2 U]\left(1 + \frac{\lambda\kappa}{S}\right)AK^{-1}G^{-1} + \frac{\lambda}{S}\left(A^2K^{-2}\Gamma^2 U + \frac{\lambda r}{S}A\right)K^{-1}G^{-1}.
\tag{153}
$$

**Remark.** *Because $\Gamma$ may not be positive semidefinite, the test loss can be larger than the training loss, which is different from the input noise case.*

*Proof.* The matrix equation obeyed by $\Sigma$ is

$$\Sigma K + K\Sigma - \lambda K\Sigma K - \frac{\lambda}{S}A\Sigma A = \frac{\lambda}{S}\mathrm{Tr}[A\Sigma]A + \frac{\lambda}{S}\left(\mathrm{Tr}[AK^{-2}\Gamma^2 U]A + AK^{-1}\Gamma U\Gamma K^{-1}A\right),\tag{154}$$

where we use the shorthand notation $K \coloneqq A + \Gamma$. Let $[A, \Gamma] = 0$. Using the trick in Appendix E.2.2, the trace term $\mathrm{Tr}[A\Sigma]$ is calculated as

$$\mathrm{Tr}[A\Sigma] = \frac{\lambda}{S}\left(\mathrm{Tr}[AK^{-2}\Gamma^2 U]\kappa + r\right),\tag{155}$$

where $\kappa \coloneqq \frac{\mathrm{Tr}[A^2 K^{-1} G^{-1}]}{1 - \frac{\lambda}{S}\mathrm{Tr}[A^2 K^{-1} G^{-1}]}$, $r \coloneqq \frac{\mathrm{Tr}[A^3 K^{-3}\Gamma^2 G^{-1} U]}{1 - \frac{\lambda}{S}\mathrm{Tr}[A^2 K^{-1} G^{-1}]}$, and $G \coloneqq 2I_D - \lambda\left(K + \frac{1}{S}K^{-1}A^2\right)$.

The training error is

$$\begin{aligned}
L_{\mathrm{train}} &= \frac{1}{2}\mathrm{Tr}[A\Sigma_\Gamma + \Gamma\Sigma_A] \\
&= \frac{1}{2}\mathrm{Tr}[K\Sigma] + \frac{1}{2}\mathrm{Tr}[AK^{-1}\Gamma U] \\
&= \frac{\lambda}{2S}\mathrm{Tr}[AK^{-2}\Gamma^2 U]\mathrm{Tr}[AG^{-1}]\left(1 + \frac{\lambda\kappa}{S}\right) + \frac{\lambda}{2S}\left(\mathrm{Tr}[A^2 K^{-2}\Gamma^2 G^{-1} U] + \frac{\lambda r}{S}\mathrm{Tr}[AG^{-1}]\right) \\
&\quad + \frac{1}{2}\mathrm{Tr}[AK^{-1}\Gamma U].
\end{aligned}\tag{156}$$

The test loss is

$$\begin{aligned}
L_{\mathrm{test}} &= \frac{1}{2}\mathbb{E}_{\mathbf{w}}\mathbb{E}_{\mathrm{B}}\left[(\mathbf{w}^{\mathrm{T}}x - \mathbf{u}^{\mathrm{T}}x)^2\right] \\
&= \frac{1}{2}\mathbb{E}_{\mathbf{w}}\left[(\mathbf{w} - \mathbf{u})^{\mathrm{T}}A(\mathbf{w} - \mathbf{u})\right] = \frac{1}{2}\mathrm{Tr}[A\Sigma_\Gamma] \\
&= \frac{\lambda}{2S}\left(\mathrm{Tr}[AK^{-2}\Gamma^2 U]\kappa + r\right) + \frac{1}{2}\mathrm{Tr}[AK^{-2}\Gamma^2 U].
\end{aligned}\tag{157}$$

Let $A$, $\Gamma$ and $U$ commute with each other. Then,

$$\Sigma = \frac{\lambda}{S}\mathrm{Tr}[AK^{-2}\Gamma^2 U]\left(1 + \frac{\lambda\kappa}{S}\right)AK^{-1}G^{-1} + \frac{\lambda}{S}\left(A^2 K^{-2}\Gamma^2 U + \frac{\lambda r}{S}A\right)K^{-1}G^{-1}.\tag{158}$$

$\square$

### E.4.3 PROOF OF COROLLARY 1

For a 1d example, the training loss and the test loss have a simple form. We use lowercase letters for 1d cases.

**Corollary 4.** *For a 1d SGD with $L_2$ regularization, the training loss and the test loss are*

$$L_{\mathrm{train}} = \frac{a\gamma}{2(a+\gamma)}\frac{2(a+\gamma) - \lambda\left[(a+\gamma)^2 + \frac{2}{S}a(a-\gamma)\right]}{2(a+\gamma) - \lambda\left[(a+\gamma)^2 + \frac{2}{S}a^2\right]}u^2,\tag{159}$$

$$L_{\mathrm{test}} = \frac{a\gamma^2}{2(a+\gamma)}\frac{2 - \lambda(a+\gamma)}{2(a+\gamma) - \lambda\left[(a+\gamma)^2 + \frac{2}{S}a^2\right]}u^2.\tag{160}$$

*Proof.* The training error and the test loss for 1d cases can be easily obtained from Theorem E.4.2. $\square$

Now we prove Corollary 1.

*Proof.* The condition for convergence is $1 - \frac{\lambda}{S}\mathrm{Tr}[A^2 K^{-1} G^{-1}] > 0$. Specifically,

$$\lambda(a+\gamma)^2 - 2(a+\gamma) + \lambda\frac{2}{S}a^2 < 0.\tag{161}$$

For a given $\gamma$, the learning rate needs to satisfy

$$\lambda < \frac{2(a+\gamma)}{(a+\gamma)^2 + \frac{2}{S}a^2}. \tag{162}$$

For a given $\lambda$, $\gamma$ needs to satisfy

$$\frac{1 - a\lambda - \sqrt{1 - \frac{2}{S}a^2\lambda^2}}{\lambda} < \gamma < \frac{1 - a\lambda + \sqrt{1 - \frac{2}{S}a^2\lambda^2}}{\lambda}, \tag{163}$$

which indicates a constraint on $\lambda$:

$$a\lambda < \sqrt{\frac{S}{2}}. \tag{164}$$

If $\gamma$ is allowed to be non-negative, the optimal value can only be $0$ due to the convergence condition. Therefore, a negative optimal $\gamma$ requires an upper bound on it being negative, namely

$$\frac{1 - a\lambda + \sqrt{1 - \frac{2}{S}a^2\lambda^2}}{\lambda} < 0. \tag{165}$$

Solving it, we have

$$a\lambda > \frac{2}{1 + \frac{2}{S}}. \tag{166}$$

By combining with Eq. (164), a necessary condition for the existence of a negative optimal $\gamma$ is

$$\frac{2}{1 + \frac{2}{S}} < \sqrt{\frac{S}{2}} \to (S-2)^2 > 0 \to S \neq 2. \tag{167}$$

Hence, a negative optimal $\gamma$ exists, if and only if

$$\frac{2}{1 + \frac{2}{S}} < a\lambda < \sqrt{\frac{S}{2}}, \text{ and } S \neq 2. \tag{168}$$

$\square$

For higher dimension with $\Gamma = \gamma I_D$, it is possible to calculate the optimal $\gamma$ for minimizing the test loss (11) as well. Specifically, the condition is given by

$$\frac{d}{d\gamma}L_{\text{test}} := \frac{1}{2}\frac{d}{d\gamma}\frac{f(\gamma)}{g(\gamma)} = 0, \tag{169}$$

where

$$f(\gamma) := \gamma^2 \text{Tr}\left[AK^{-2}\left(I_D + \frac{\lambda}{S}A^2K^{-1}G^{-1}\right)U\right], \tag{170}$$

$$g(\gamma) := 1 - \frac{\lambda}{S}\text{Tr}[A^2K^{-1}G^{-1}]. \tag{171}$$

Although it is impossible to solve the equation analytically, it can be solved numerically.

