# OpenReview forum: "Strength of Minibatch Noise in SGD"
_ICLR.cc/2022/Conference — ICLR 2022 Spotlight_

### Official Review · Reviewer_5twY · 2021-11-01

**Correctness:** 3
**Technical Novelty And Significance:** 3
**Empirical Novelty And Significance:** 3
**Recommendation:** 8
**Confidence:** 3

**Main Review:**

Pros:
The structure of the paper is well-orgainzed. The problem is well-motived with proper assumptions. The proofs look clear.  The mentioned applications in the paper are practical and novel, it is interesting to see this work could promote the understanding of many empirical findings.
Cons:
I have some concerns as following list though.
1. in Figure 4, the proposed theory saw a drop around $\lambda~1.7$, can you explain this phenomenon and what does it mean when $\lambda>1.7$. same question for Figure 3 right and Figure 5 where exact solution has a sudden drop/rise.
2. in equation 53, the first equality replies on approximating $(l_i')^2$ by average. this could be an issue if there exist some relationship between $l_i'$ and $\nabla f$, for instance $f$ is quadratic function.
3. in equation 54, the sign of $\Gamma w^\star$ should be positive? Is this a typo?
4. in equation 55, how do we get $\mathcal{O}(S^{-2})$ term for the first equality? same for equation 57
5. in equation 117, why there's $t-1$ subscript for $w$ after first equality?
6. line spacing needs fix: equation 131

**Summary Of The Paper:**

This paper studies the minibatch noise for discrete time SGD and discusses its approximation on different applications. The novelty of the paper stands on the derivation of minibatch noise covatiance of discrete time SGD. For special cases with label noise and L2 regularization, this work derives the exact solution of covariance. For a more general setting, the author gives the covariance replying on two assumptions. In application section, a few experiments are presented to show the strength of the theory.

**Summary Of The Review:**

This paper finds the novelty of discrete sgd noise. Compared with previous with Hessian noise approximation, this paper found the shape of sgd noise on a generic setting. Experiments shows much better consistency with this work. Throughout the paper, the theoretical derivations are solid and numerical experiments are well-design. Suggest acception.

---

> ### Author Response · Authors · 2021-11-14
> **Reply**
>
> *Q1: A sudden drop/rise can be seen in Figures 3 and 4.  Can you explain this phenomenon and what does it mean?*
>
> * Sorry for the ambiguity. It is not a drop/rise but a straight vertical line that shows where the theory predicts a divergence, and it can be seen from the plot that this agrees with where the numerical results show divergence. We added a description of this fact in the manuscript to improve the description of the figures.
>
> *Q2: In equation 53, the first equality replies on approximating $(l_i’)^2$ by average. This could be an issue if there exist some relationship between $l_i’$ and $\nabla f$ , for instance f is quadratic function.*
>
> * Sorry for the inaccuracy in our original statement in the proof. This sentence you referred to (“by performing an approximation...”) should be replaced by (a more accurate statement): “By assumption 2,” and Eq. 53 directly follows from the assumption without any further approximation. In fact, we believe that it would be an important future step towards understanding SGD by relaxing assumption 2.
>
> *Q3: in equation 54, the sign of $\Gamma w^*$ should be positive? Is this a typo?*
>
> * Thank you for pointing out this typo. Yes, this is indeed a typo, and the sign should be positive. We checked the rest of the proof, and other results remain unchanged.
>
>
> *Q4: in equation 55, how do we get $O(S^{-2})$ term for the first equality? same for equation 57?*
>
> * This is because the expectation of $(w-w^*)(w-w^*)^T$ is $\Sigma$ (by definition), which is of $O(S^{-1})$ by assumption 1. Therefore, $\frac{1}{S}\Sigma$ is of order $O(S^{-2})$.
>
> *Q5: in equation 117, why there's t-1 subscript for w after first equality?*
>
> * We apologize for causing confusion. Keep in mind that we are considering the asymptotic regime where the stationary distribution exists, and so the noise is independent of time (by stationarity), and so the subscript $t-1$ can be neglected.
>
>
> *Q6: line spacing needs fix: equation 131.*
>
> * Thanks for pointing it out. We will fix it.

---

### Official Review · Reviewer_pu11 · 2021-11-02

**Correctness:** 4
**Technical Novelty And Significance:** 4
**Empirical Novelty And Significance:** Not applicable
**Recommendation:** 8
**Confidence:** 3

**Main Review:**

Strengths:
- The paper is very well-written and easy to read.
- It provides clear explanations on the theorems and formulae with breakdown of the expression.
- The results are novel and close a few important gaps.
- Implications discussed in Section 6 are particularly interesting and match the empirical observations with the new theory.
- To the extent I have checked the theorems are sound and derivations are accurate.

Minor comments:
- Pg. 2: The following should be stated as an assumption that might not hold in practice. “One can decompose the gradient into a deterministic plus a stochastic term.”
- Pg. 3: The following sentence would not hold for loss functions without a global minimum at zero: “This proposition implies that there is no noise if our model can achieve zero training loss (which is achievable for an overparametrized model)”.
- Pg. 5: “By definition, C = J is the FIM”. J is used without proper definition.
- Pg. 8: Is there a typo in this sentence: “Figure 1-Right confirms that, at a large learning rate, the optimal weight decay can indeed be optimal.”
- Could your analysis be extended to handle learning rate schedules such as step decay? For example, it has been observed that the gradient variance can increase for a while after a learning rate drop in common neural network training [1]. That would not exactly match Theorem 5 that predicts shrinking covariance by the loss. Any thoughts?

[1] Faghri, F., Duvenaud, D., Fleet, D. J., & Ba, J. (2020). A Study of Gradient Variance in Deep Learning. arXiv preprint arXiv:2007.04532.


**Summary Of The Paper:**

This paper studies the properties of the gradient noise in mini-batch SGD using discrete-time analysis for a fixed learning rate. Their analysis is more general than prior work and considers SGD with momentum, a learning rate matrix (that subsumes preconditioning methods), and regularization. They provide closed form expressions for the noise covariance for various machine learning problems including linear regression with label noise (Theorem 3), with regularization (Proposition 3), and more general models under reasonable assumptions (Theorem 5). Their theory matches the empirical observations about the effect of mini-batch size and learning rate on the gradient noise for small batch sizes and large learning rates where there was previously a gap in theory. Section 6 illustrates the significance of the results as it is able to explain various observations in machine learning theory and deep learning practice as well as providing ideas for practitioners.

**Summary Of The Review:**

This work makes a clear contribution and is very well written.

---

> ### Author Response · Authors · 2021-11-14
> **Reply**
>
> *Q1: Pg. 2: The following should be stated as an assumption that might not hold in practice. “One can decompose the gradient into a deterministic plus a stochastic term.”*
>
> * We are referring to the following fact. For any random variable $x$, one can decompose $x$ as $x=\mathbb{E}[x]+\delta$, where $\delta$ is a zero-mean random variable. Therefore, this sentence is not an assumption but a fact.
>
>
> *Q2: Pg. 3: The following sentence would not hold for loss functions without a global minimum at zero: “This proposition implies that there is no noise if our model can achieve zero training loss (which is achievable for an overparametrized model)”.*
>
> * Thanks for pointing this out. Here, we are (implicitly) assuming that the loss function is shifted such that the global minimum has zero loss. We will make this explicit.
>
>
> *Q3: Pg. 8: Is there a typo in this sentence: “Figure 1-Right confirms that, at a large learning rate, the optimal weight decay can indeed be optimal.”*
> * Thanks for pointing it out. It is indeed a typo. The correct sentence should be “Figure 1-Right confirms that, at a large learning rate, the optimal weight decay can indeed be negative.”
>
> *Q4: Could your analysis be extended to handle learning rate schedules such as step decay? For example, it has been observed that the gradient variance can increase for a while after a learning rate drop in common neural network training [1]. That would not exactly match Theorem 5 that predicts shrinking covariance by the loss. Any thoughts?*
> * In principle, Eq. (14) in theorem 5 can be applied to non-asymptotic situations such as the case you described, where the learning rate is suddenly decreased (even regimes though the rest of the paper is focused on the asymptotic time regime). As we have commented in the conclusion of the paper, this regime is a very important future work to pursue.
>
> * Also, the paper you pointed to is indeed a very curious work, and it should be interesting to apply our result to see whether such a sudden increase of variance follows a learning rate drop. As a side note, our preliminary calculation shows that Eq. (14) (combined with proper calculations of the non-asymptotic fluctuation) may indeed suggest a rise in variance as the learning rate is dropped -- but this is beyond the scope of the present work, and we will perhaps formally study this curious phenomenon in future work.

---

> > ### Comment · Reviewer_pu11 · 2021-11-21
> > **Thank you**
> >
> > I thank the authors for their response to my review. I encouraged the authors to incorporate their responses. I keep my recommendation for acceptance.

---

### Official Review · Reviewer_4YZA · 2021-11-03

**Correctness:** 4
**Technical Novelty And Significance:** 3
**Empirical Novelty And Significance:** 2
**Recommendation:** 8
**Confidence:** 3

**Main Review:**

Strengths:
- Clarity: The paper is well written, and the main technical results are well presented.

- Novelty: Understanding the generalization capability of SGD is an active research topic. The paper proposes a slightly different approach than previous works, finding an analytical solution to the noise shape of SGD in simple scenario rather than relying on continuous-time or Hessian-based approximation. This approach allows to highlight the importance of some factor, such as the loss level, that were neglected in previous works. I think this finding would be of interest to the community.

- Significance: The analysis also provides some insight on more applicative issue. It highlights why the linear scaling of the learning rate might not work for small batches or large learning rate. It also provides justification for the use of negative weight decay or insight about stability in second order approaches.

Weakness:
- The analysis of the noise structure in the generic setting relies on two assumptions and it is unclear if those assumptions are realistic in practice, in particular assumption 2 about loss homogeneity.


**Summary Of The Paper:**

This paper investigates the importance of noise in mini-batch SGD. The main contribution of this paper is to derive an analytic solution to the shape and strength of SGD minibatch noise for linear regression with random noise in the label, linear regression with additional L2 regularization and non-linear regression given some assumption about the model fluctuation and loss homogeneity.

Those analysis reveal that 1) the SGD noise is proportional to the loss level 2) the shape of the noise differs for different loss minima. This is in contrast with previous work looking at this problem using various approximation which neglects those effect.

Authors finally highlight various applications for those findings. In particular, the analytical formulation they derive for the noise explains why the linear scaling rule of the learning rate does not work for small batch or high learning rate.


**Summary Of The Review:**


This paper investigates the role minibatch SGD noise from a different perspective and provide novel insight about the strength and the shape of the noise. Paper finding could be of interest to the community, I therefore recommend acceptance.

---

> ### Author Response · Authors · 2021-11-14
> **Reply**
>
> *Q1: “The analysis of the noise structure in the generic setting relies on two assumptions and it is unclear if those assumptions are realistic in practice, in particular assumption 2 about loss homogeneity.”*
>
> * Thanks for pointing this out. We agree that this is a limitation of the proposed theory, and, as mentioned in the manuscript, we think it is an important future step to relax this assumption. We also point out that one can replace the second assumption with the decoupling assumption/approximation, which has also been proposed in Mori et al. (2021) and Wojtowytsch (2021) (see the last remark on page 6). The decoupling approximation, in turn, has been tested empirically for realistic neural network settings in Mori et al. (2021), and so we believe our result, even if not based on the loss homogeneity assumption, is still relevant for machine learning and deep learning.

---

### Official Review · Reviewer_oBNf · 2021-11-04

**Correctness:** 3
**Technical Novelty And Significance:** 3
**Empirical Novelty And Significance:** 1
**Recommendation:** 6
**Confidence:** 3

**Main Review:**

**Strengths:**
- Notation is clear and the background nicely sets up the distinction between the noise $C$ and the fluctuation $\Sigma$ as the two central objects of study in this work
- I like where you discuss the "crucial messages" a theoretical claim delivers.
- I think your thought experiments considering interpolating regimes as a means of demonstrating a more carful understanding of noise (beyond crude approximations) is needed are very good.

**Weaknesses:**
- The major weakness of this work is while you clearly explain how your theory differs from previous works and introduces fewer "approximations", I keep coming back to a line in your introduction, "the limitation of these approximations is not well understood".   So, I would expect to not just understand how your theory differs, but be clearly explained and demonstrated (empirically) where the limitations in the previous approximations are and how your theory provides explanation or avoids these limitations.  I feel that many of the experiments in section A of your appendix are very valuable to demonstrate this and I would pull these up to the main and discuss directly.
- Section 6 applications discuss a broad range of claims and connections to your theory, but most of them are not supported empirically (such as the connection to second order methods, $\lambda - S$ scaling law, high dimensional regression) and its left to the reader to assume that expressions you give explain the limitations in the previous approximations.  Again if the limitations are not well understood how should we evaluate the more complex expressions you derive in this work.  For example, in section 6.5 Failure of $\lambda - S$ scaling law, you write "it is known that this scaling law fails when the learning rate is too large, or the batch size is too small", but you do not provide empirical evidence or citation of this fact.  You then state that "our result in Theorem 2 suggests the reason for the failure" and demonstrate that only "the leading term is indeed proportional to $\lambda/S$" but you don't demonstrate that including the higher order terms corroborates the evidence demonstrating how the scaling law fails.  I think spending section 6 to focus deeply on a few applications highlighting your theory would be more valuable than trying to address a broad range of applications shallowly.
- In section 4.3, it is not clear to me how "[L2] regularization also causes a unique SGD noise".  How could regularization introduce gradient noise given that the gradient for the regularization term is independent of the batch or label and is a deterministic function of the parameters?   Indeed considering the expression for $\eta_t = \frac{1}{S} \Sigma_{i \in B_t} \nabla \ell (x_i,y_i,w_{t-1}) - \mathbb{R}_B[\hat{g}_t]$ given in background it seems clear that this term would be independent of $\gamma$. I can understand how regularization could effect the fluctuation $\Sigma$, but not how it could change the noise $C$.  Please explain the sentence "this term is due to the mismatch between the regularization and the minimum of the original loss" in more detail.
- Throughout your work you borrow heavily upon the theory developed in Liu et al. 2021 (i.e. it seems that almost all your theory builds upon Theorem 1 which is from their work), but you don't address this work directly in related work and scatter throughout the body references to how your expressions differs from theirs.  I would devote a whole paragraph either in related work or discussion to discuss how your work builds up or overlaps with their work.

**Minor comments:**
- You write in section 4.2 "a Hessian approximation fails to account for the randomness in the data of strength $\sigma^2$, but why wouldn't the  $\sigma^2$ be absorbed into the scalar constant $c_0$ in the Hessian approximation?  Its not clear to me that this is a setting that is a limitation for the Hessian approximation.
- Consider discussing the recent empirical work "Stochastic Training is Not Necessary for Generalization"

**Summary Of The Paper:**

This work studies the noise (covariance of gradients) and fluctuation (covariance of parameter distribution) in the limiting setting of linear regression and for a general loss trained with discrete SGD.

**Summary Of The Review:**

In summary, I think this is a good paper and leaning towards accept.  I think if the authors address some of the weakness discussed above then I will raise my score.

---

> ### Author Response · Authors · 2021-11-14
> **Reply**
>
> *Q1: "The major weakness of this work is while you clearly explain how your theory differs from previous works and introduces fewer "approximations", I keep coming back to a line in your introduction, "the limitation of these approximations is not well understood". So, I would expect to not just understand how your theory differs, but be clearly explained and demonstrated (empirically) where the limitations in the previous approximations are and how your theory provides explanation or avoids these limitations. I feel that many of the experiments in section A of your appendix are very valuable to demonstrate this and I would pull these up to the main and discuss directly."*
>
> * Thanks for suggesting that the experiments in section A can be moved to the main text to improve the main text. We agree to this. However, we had to put Section A in the appendix due to (1) the fact that this comparison is quite technical and (2) space constraints. We added more detailed discussions in section A and an explicit reference to Section A in the main text.
>
>
> *Q2: "Section 6 applications discuss a broad range of claims and connections to your theory, but most of them are not supported empirically (such as the connection to second order methods,  scaling law, high dimensional regression) and its left to the reader to assume that expressions you give explain the limitations in the previous approximations. Again if the limitations are not well understood how should we evaluate the more complex expressions you derive in this work. For example, in section 6.5 Failure of  scaling law, you write "it is known that this scaling law fails when the learning rate is too large, or the batch size is too small", but you do not provide empirical evidence or citation of this fact. You then state that "our result in Theorem 2 suggests the reason for the failure" and demonstrate that only "the leading term is indeed proportional to " but you don't demonstrate that including the higher order terms corroborates the evidence demonstrating how the scaling law fails."*
>
> * Thanks for this comment. Let us first point out a factual error in your criticism. For the case of high dimensional linear regression, we performed careful numerical results to compare with the predictions of proposition 4 and theorem 6. This numerical experiment is presented in sections A.2 and Figure 6. The theory is observed to agree well with the experiment. We now add a direct reference to this experiment in the main text.
>
> * For the failure of the $\lambda-S$ scaling law, we did miss a reference here. One such piece of evidence can be found in the famous paper of: https://arxiv.org/pdf/1706.02677.pdf. For example, in Figure 1 of this work, the generalization error of the model on Imagenet is invariant for a batch size range from 64 to 8000. However, such invariance breaks down from batch size 8000-64000. We have added this reference to the paper.
>
>
> *Q3: "In section 4.3, it is not clear to me how "[L2] regularization also causes a unique SGD noise". How could regularization introduce gradient noise given that the gradient for the regularization term is independent of the batch or label and is a deterministic function of the parameters? Indeed considering the expression … given in background it seems clear that this term would be independent of . I can understand how regularization could effect the fluctuation , but not how it could change the noise . Please explain the sentence "this term is due to the mismatch between the regularization and the minimum of the original loss" in more detail."*
>
> * Thanks for this important question. One crucial feature of the actual minibatch noise is that it depends on the parameter $w$, namely, $C(w)$ is a function of the parameter (because the gradient is, in general, a function of $w$). While $C$ does not directly depend on $\gamma$, the learned parameter $w$ depends non-trivially on $\gamma$. Therefore, the actual noise covariance indirectly depends on $\gamma$ through dependence on $w$.
>
> * The sentence "this term is due to the mismatch between the regularization and the minimum of the original loss" refers to the following (relatively subtle) insight: the L2 regularization and the data-dependent loss can be seen as two different losses with different local minima. In this perspective, the L2 regularization has only one minimum, namely, $w=0$. In contrast, the data-dependent loss, in general, has minima that are not $w=0$; hence there is a mismatch between the location of the data-dependent loss and that of the L2 regularization. One can show that if there is no mismatch between the local minima of the two parts of the loss, then the L2 regularization does not cause a noise asymptotically. In our result, this mismatch is captured by the matrix $U:= uu^T$. When $U=0$ (namely, when the minimum of the data-dependent loss is $w=u=0$), then this part of the noise vanishes.

---

> > ### Author Response · Authors · 2021-11-14
> > **Reply part 2**
> >
> >
> > *Q4: "Throughout your work you borrow heavily upon the theory developed in Liu et al. 2021 (i.e. it seems that almost all your theory builds upon Theorem 1 which is from their work), but you don't address this work directly in related work and scatter throughout the body references to how your expressions differs from theirs. I would devote a whole paragraph either in related work or discussion to discuss how your work builds up or overlaps with their work."*
> >
> > * Thanks for pointing this out. The crucial difference is that Liu et al. (2021) only considers an artificially injected noise, with an assumed noise covariance matrix (either proportional to the identity or the Hessian), while this work considers the actual minibatch noise that is caused by the random selection of data samples during training and, therefore, every noise covariance needs to be respectively derived in this work -- and this is a technical advancement of this work over Liu et al. (2021). Our work is thus also more relevant to the actual deep learning practice of training with SGD. This difference has been made clear in Table 1.

---

### Author Response · Authors · 2021-11-14
**Rebuttal Summary**

We thank the reviewers for supporting this work and agreeing on its academic value: a systematic study of the noise and fluctuation of the discrete-time minibatch SGD, with many possible theoretical or practical machine learning applications. We reply to the constructive questions of the reviewers below.

---

### Decision · Program_Chairs · 2022-01-20

**Decision:**

Accept (Spotlight)

**Comment:**

All the reviewers think that the work is significant and new. Therefore, they support the paper to be published at ICLR 2022. Given the strong results and the “accept” consensus from the reviewers, I accept the paper as “spotlight”. The authors should implement all the reviewers’ suggestions into the final version.